# SecEmb: Sparsity-Aware Secure Federated Learning of On-Device Recommender System with Large Embedding

**Peihua Mai** [1] [*]    **Youlong Ding** [2] [*]    **Ziyan Lyu** [3]    **Minxin Du** [4]    **Yan Pang** [1]

## Abstract

Federated recommender system (FedRec) has emerged as a solution to protect user data through collaborative training techniques. A typical FedRec involves transmitting the full model and entire weight updates between edge devices and the server, causing significant burdens to devices with limited bandwidth and computational power. While the sparsity of embedding updates provides opportunity for payload optimization, existing sparsity-aware federated protocols generally sacrifice privacy for efficiency. A key challenge in designing a secure sparsity-aware efficient protocol is to protect the rated item indices from the server. In this paper, we propose a lossless secure recommender systems on sparse embedding updates (SecEmb). SecEmb reduces user payload while ensuring that the server learns no information about both rated item indices and individual updates except the aggregated model. The protocol consists of two correlated modules: (1) a privacy-preserving embedding retrieval module that allows users to download relevant embeddings from the server, and (2) an update aggregation module that securely aggregates updates at the server. Empirical analysis demonstrates that SecEmb reduces both download and upload communication costs by up to 90x and decreases user-side computation time by up to 70x compared with secure FedRec protocols. Additionally, it offers non-negligible utility advantages compared with lossy message compression methods.

---

[*]Equal contribution  [1]National University of Singapore, Singapore [2]Hebrew University of Jerusalem, Jerusalem, Israel [3]NUS (Chongqing) Research Institute, Chongqing, China [4]Hong Kong Polytechnic University, Hong Kong SAR, China. Correspondence to: Yan Pang <jamespang@nus.edu.sg>.

*Proceedings of the $42^{nd}$ International Conference on Machine Learning*, Vancouver, Canada. PMLR 267, 2025. Copyright 2025 by the author(s).

## 1. Introduction

Personalized recommendation systems (RecSys) model the interactions between users and items to uncover users' interests. To understand the underlying preferences of users and properties of items, various model-based approaches have been developed to learn hidden representations of both users and items (Koren et al., 2009; Xue et al., 2017; Rendle, 2010). These methods embed users and items into fixed-size latent vectors, which are then used to predict interactions. The parameters of these latent vectors are known as user and item embeddings.

The development of personalized recommendation systems (RecSys) relies heavily on collecting user profiles and behavioral data, such as gender, age, and item interactions. However, the sensitive nature of this information often makes users hesitate to share it with service providers. Recent advancements in edge computing have provided a potential solution through federated learning (FL), which allows users to collaboratively train models on their local devices without exposing personal data (McMahan et al., 2017). In a typical federated recommender system (FedRec), edge devices download the model from the server, perform local training, and upload weight updates for aggregation. During update aggregation, plaintext gradients can reveal user information. To mitigate this risk, secure aggregation (SecAgg) (Bonawitz et al., 2017) is adopted to prevent the server from inspecting individual updates during training.

Despite its effectiveness in preserving privacy, the above paradigm suffers significant communication and computation overheads as it requires transmission of the full model and complete updates (Bonawitz et al., 2019). This issue becomes particularly problematic in latent factor-based RecSys, where the payload scales linearly with the number of items, potentially leading to substantial burden on resource-constrained edge devices with: (1) limited communication bandwidth, as the communication bandwidth between edge devices and the central server is often constrained; and (2) limited user computational power and storage, since edge devices generally have limited processing capabilities, memory, and storage compared to centralized servers.

Fortunately, users typically interact with only a small sub-

set of available items in practice, which presents two key opportunities for payload optimization. First, only the embeddings of the items a user interacts with are relevant for model updates. As a result, users can retrieve and store only these relevant item embeddings, significantly reducing computational and storage overhead. Secondly, the update vector is highly sparse in that only a small subset of the item embeddings are non-zero. Therefore, it is desirable to make the per-user communication and computation succinct, i.e., independent of or logarithmic in the item size.

A key challenge in designing sparsity-aware efficient FedRec is to ensure that the *indices or coordinates of non-zero elements* remain hidden from the server. Existing sparsity-aware FL protocols reduce communication overhead at the cost of increased privacy leakage (Lin et al., 2022; Liu et al., 2023; Lu et al., 2023), as they inevitably reveal the coordinates of non-zero elements or significantly narrow the set of potential non-zero updates. Such coordinate information is particularly sensitive in RecSys, as it directly indicates which items a user has rated. This raises a critical question: *Can we achieve succinct user communication and computation cost in FedRec, while ensuring that the server learns no information about both the rated item indices and user updates except the aggregated model?*

In this paper, we address the problem by proposing a sparsity-aware secure recommender systems with large embedding updates (SecEmb). The implementation is available at https://github.com/NusIoraPrivacy/SecEmb. Our SecEmb consists of two correlated modules: (1) a privacy-preserving embedding retrieval module that allows users to download relevant embeddings from the server, and (2) an update aggregation module that securely aggregates updates at the server. Our protocol is efficient, secure, and lossless:

- **Efficient edge device update.** SecEmb achieves succinct download and upload communication, with optimized user computation and memory costs as operations are performed only on rated item embeddings.
- **Privacy-preserving model training.** Both rated item indices and user updates (including non-zero embedding index and their gradient values) remain hidden from the server throughout the training process.
- **Lossless message compression**. Unlike dimension reduction or quantization methods, SecEmb reduces communication costs without compromising accuracy.

The contribution of our work can be summarized as follows:

(1) Leveraging the sparsity of embedding updates, we develop a lossless efficient FedRec training protocol that achieves succinct user communication and computation costs, while ensuring the privacy of individual updates.

(2) We further optimize the payload of SecEmb by exploiting the row-wise sparsity of the embedding update matrix as well as the correlation between privacy-preserving embedding retrieval and update aggregation modules.

(3) Empirical studies demonstrate that SecEmb reduces both download and upload communication costs by up to 90x and decreases user-side computation time by up to 70x compared with FedRec utilizing the most efficient SecAgg protocol, with non-negligible utility advantages over lossy message compression schemes.

## 2. Related Work

### 2.1. Cross-User Federated Recommender System

In recent years, federated recommender system (FedRec) trained on individual users has gained growing interest in research community. FCF (Ammad-Ud-Din et al., 2019) and FedRec (Lin et al., 2020) are among the pioneering implementations of federated learning for collaborative filtering based on matrix factorization. Privacy guarantees are enhanced through the application of cryptographic methods to the transmitted gradients (Chai et al., 2020; Mai & Pang, 2023). FedMF (Chai et al., 2020) ensures privacy with homomorphic encryption (HE) techniques, while incurring substantial computation overhead. LightFR (Zhang et al., 2023) sacrifices utility for efficiency by binarizing continuous user/item embeddings through learning-to-hash. Difacto (Li et al., 2016) introduces a distributed factorization machine algorithm that is scalable to a large number of users and items. FedNCF (Perifanis & Efraimidis, 2022) is a federated realization of neural collaborative iltering (NCF), where secure aggregation is leveraged to protect user gradients. FMSS (Lin et al., 2022) proposes a federated recommendation framework for several recommendation algorithms based on factorization and deep learning. (Rabbani et al., 2023) and (Xu et al., 2022) improve the training efficiency for edge device using locality-sensitive hashing (LSH) techniques (Chen et al., 2020; 2019). Despite the development of various algorithms for FedRec systems, there is a lack of research on designing sparsity-aware efficient FedRec while simultaneously ensure privacy and utility.

### 2.2. Secure Aggregation for Machine Learning

Secure Aggregation (SecAgg) computes the summation of private gradients without revealing any individual updates. (Bonawitz et al., 2017) introduces a secure aggregation protocol for FL, leveraging a combination of pairwise masking, Shamir's Secret Sharing, and symmetric encryption techniques. (Bell et al., 2020) reduces the communication and computation overhead to depend logarithmic in the client size. FastSecAgg (Kadhe et al., 2020) designs a multi-secret sharing protocol based on Fast Fourier Transform to save computation cost. SAFELearn (Fereidooni et al., 2021) designs an secure two-party computation protocol for efficient

FL implementation. LightSecAgg (So et al., 2022) reduces the computation complexity via one-shot reconstruction of aggregated mask. The two-server additive secret sharing (ASS) protocol (Xiong et al., 2020) represents the most efficient SecAgg approach in terms of computation and communication complexity. Refer to Table 6 for the complexity of existing SecAgg algorithms. However, current SecAgg protocols incur communication costs that scale linearly with model size, and existing attempts on sparse update aggregation (Ergun et al., 2021; Liu et al., 2023; Lu et al., 2023) fail to ensure security for individual updates (including both indices and values of non-zero updates).

# 3. Background and Preliminaries

## 3.1. Problem Statement

In FedRec, a number of users want to jointly train a recommendation system based on their private data. Denote $\mathcal{U} = \{u_1, u_2, ..., u_n\}$ as the set of users and $\mathcal{I} = \{i_1, i_2, ..., i_m\}$ as the set of available items. Each user $u \in \mathcal{U}$ has a private interaction set $\mathcal{R}_u = \{(i, r_{u,i}) | i \in \mathcal{I}_u\} \subset [m] \times \mathbb{R}$, where $\mathcal{I}_u$ denotes the set of items rated by user $u$ and $r_{u,i}$ denotes the rating user $u$ gives to item $i$. Denote $X \in \mathbb{R}^{n \times l_x}$ and $Y \in \mathbb{R}^{m \times l_y}$ as the user and item feature matrix, respectively, capturing user and item attributes such as demographic details, genre, or price. Note that $\mathcal{R}_u$ reflects user-item interactions, excluding these auxiliary features. Our goal is to generate a rating prediction that minimizes the squared deviation between actual and estimated ratings.

We focus on a class of RecSys that models low-dimensional latent factors for user and items (Koren et al., 2009; Xue et al., 2017; Rendle, 2010). The recommender fits a model $f$ comprising of $d$-dimensional latent factors (or embeddings) for user $P \in \mathbb{R}^{n \times d}$ and item $Q \in \mathbb{R}^{m \times d}$, along with the remaining parameters $\theta$. Denote $p_u \in \mathbb{R}^d$ and $q_i \in \mathbb{R}^d$ as the latent factors (or embeddings) for user $u$ and item $i$. A general form of the rating prediction can be expressed as:

$$\hat{r}_{u,i} = f(x_u, y_i; p_u, q_i, \theta), \tag{1}$$

where $x_u \in \mathbb{R}^{l_x}$ and $y_i \in \mathbb{R}^{l_y}$ denote the feature vector for user $u$ and item $i$, and $\hat{r}_{u,i}$ is the estimated prediction for user $u$ on item $i$.

Denote $l(\cdot)$ as a general loss function. The model is trained by minimizing:

$$\mathcal{L} = \sum_{u,i} l(r_{u,i}, \hat{r}_{u,i}) \tag{2}$$

The remaining parameters $\theta$ typically include but are not limited to: (1) Feature extractors that convert user and item feature vectors into fixed size representations, denoted as $F_x : \mathbb{R}^{l_x} \to \mathbb{R}^{l_x \times d}$ and $F_y : \mathbb{R}^{l_y} \to \mathbb{R}^{l_y \times d}$, respectively.

(2) The feed-forward layers within a deep neural network model.

In each training round, users locally update their private parameters $\Theta_p$ and upload their updates of public parameters $\mathbf{g}_{\Theta_s}$ to the server. To safeguard the privacy of individual gradients, the server employs SecAgg to aggregate the gradients from all active clients and update the public model $\Theta_s$.

## 3.2. Function Secret Sharing

Our protocol builds on function secret sharing (FSS) (Boyle et al., 2015; 2016) to optimize the communication payload. FSS secret shares a function $f : \{0, 1\}^n \to \mathbb{G}$, for some abelian group $\mathbb{G}$, into two functions $f_1$, $f_2$ such that: (1) $f(x) = \sum_{i=1}^{2} f_i(x)$ for any $x$, and (2) each description of $f_i$ hides $f$.

**Definition 3.1** (Function Secret Sharing). A function secret sharing (FSS) scheme with respect to a function class $\mathcal{F}$ is a pair of efficient algorithms (FSS.Gen, FSS.Eval):

- FSS.Gen($1^\lambda$, $f$): Based on the security parameter $1^\lambda$ and function description $f$, the key generation algorithm outputs a pair of keys, $(k_1, k_2)$.
- FSS.Eval($k_i$, $x$): Based on key $k_i$ and input $x \in \{0, 1\}^n$, the evaluation algorithm outputs party $i$'s share of $f(x)$, denoted as $f_i(x)$. $f_1(x)$ and $f_2(x)$ form additive shares of $f(x)$.

FSS scheme should satisfy the following informal properties (defined formally in Appendix B.2):

- **Correctness:** Given keys $(k_1, k_2)$ of a function $f \in \mathcal{F}$, it holds that FSS.Eval($k_1, x$) + FSS.Eval($k_2, x$) = $f(x)$ for any $x$.
- **Security:** Given keys $(k_1, k_2)$ of a function $f \in \mathcal{F}$, a computationally-bounded adversary that learns either $k_1$ or $k_2$ gains no information about the function $f$, except that $f \in \mathcal{F}$.

A naive form of FSS scheme is to additively secret share each entry in the truth-table of $f$. However, this approach results in each share containing $2^n$ elements. To obtain polynomial share size, nontrivial scheme of FSS has been developed for simple function classses, e.g., point functions (Boyle et al., 2015; 2016). Our approach utilizes the advanced FSS scheme for the point function. In the following, we provide the formal definition of point function.

**Definition 3.2** (Point Function). For $\alpha \in \{0, 1\}^n$ and $\beta \in \mathbb{G}$, the point function $f_{\alpha,\beta} : \{0, 1\}^n \to \mathbb{G}$ is defined as $f_{\alpha,\beta}(\alpha) = \beta$ and $f_{\alpha,\beta}(x) = 0$ for $x \neq \alpha$.

# 4. Methodology

Observing that the public gradients primarily consist of highly sparse item embedding updates, we propose SecEmb, a secure FedRec protocol optimized to reduce the costs associated with item embeddings. Figure 1 illustrates the design of SecEmb, which is comprised of two modules: (1) a privacy-preserving embedding retrieval module and (2) an update aggregation module.

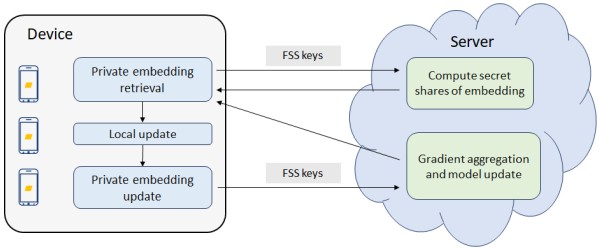

*Figure 1.* Overview of SecEmb, which is comprised of two secure and communication-efficient modules: (1) A secure embedding retrieval module and (2) A secure embedding aggregation module.

## 4.1. Key observation

In most practical recommendation scenarios, the number of items a user has previously interacted with is typically much smaller than the total number of available items (see Figure 2(a)). This observation is linked to the information overload phenomenon, which recommender systems aim to address. Consequently, the gradient of the item embedding is zero for all items except those with which the user has previously interacted.

Denote $\mathbf{g}_Q$ as the gradient of item embedding $Q$, which is a sparse matrix. In a typical FedRec with general-purpose SecAgg, the communication cost to download embedding $Q$ and upload the sparse matrix $\mathbf{g}_Q$ is at least $O(bmd)$, where $b$ is number of bits required to represent a single numerical value. The computation cost is at least $O(md)$. It is important to note that this corresponds to the bottleneck, since the embedding layer dominates the total model size as the item size increases (see Figure 2(b)).

Our goal is to optimize the payload for embedding layer, as this can significantly reduce the overall overhead, particularly under huge value of item size $m$.

## 4.2. Initial Construction of EmbSec

### 4.2.1. Privacy-preserving Embedding Retrieval

In FedRec, only the embeddings for interacted items are relevant for model updates. Therefore, the user can retrieve and store only these item embeddings during training instead of the entire embedding matrix. Accordingly, we design a privacy-preserving embedding retrieval protocol that allows

users to download targeted embeddings from the server without exposing the corresponding item indices.

Our key idea is to encode the rated item $\mathbf{g}_Q \in \mathbb{R}^{m \times d}$ into some point functions. Then we can construct a 2-server private retrieval scheme based on the function secret sharing (FSS) of these point functions. Suppose the user rates $m'_u$ items. The private retrieval process can be performed using the following steps:

**Step 1: Encode rated item with a point function, $\mathbf{g}_{i^u} \to f_{u,i}$.** User $u$ begins by encoding each rated item $i \in [m'_u]$ with a point function $f_{u,i} : \mathcal{I} \to \mathbb{G}$, for some abelian group $\mathbb{G}$. The function $f_{u,i}$ takes an item id $x \in \mathcal{I}$ as input and outputs $f_{u,i}(x) = 1 \in \mathbb{R}$ if $x = i$, and 0 elsewhere.

**Step 2: Generate keys for the point function.** User $u$ secret shares each function $f_{u,i}$ with FSS scheme and outputs a pair of keys, i.e., $(reK^0_{u,i}, reK^1_{u,i}) = \text{FSS.Gen}(1^\lambda, f_{u,i})$. The keys $reK^0_{u,i}$ and $reK^1_{u,i}$ are sent to server 0 and 1, respectively.

**Step 3: Compute secret shares of item embedding.** On receiving $reK^s_u$, each server $s \in \{0,1\}$ computes their secret shares of the target item embeddings as follows:

$$\mathbf{v}^s_{u,i} = \sum_j \text{FSS.Eval}(reK^s_{u,i}, j) \cdot Q_j \ \ \forall i \in [m'_u]. \quad (3)$$

**Step 4: Reconstruct embedding of target item.** On receiving $\mathbf{v}^s_{u,i}$ from two servers, user $u$ recovers the embeddings of target items by:

$$Q_{\text{idx}(i)} = \mathbf{v}^0_{u,i} + \mathbf{v}^1_{u,i} \ \ \forall i \in [m'_u], \quad (4)$$

where $\text{idx}(i)$ denotes the global index of the $i$-th rated item.

### 4.2.2. Secure Aggregation on Embedding Update

The update aggregation module also leverages FSS to reduce communication costs. Exploiting the sparsity of the embedding update matrix $\mathbf{g}_Q \in \mathbb{R}^{m \times d}$, each non-zero gradient value is encoded into some point functions. We begin with the case where user $u$ rates a single item, i.e., $m'_u = 1$.

Suppose user $u$'s item embedding update lies in a sparse gradient matrix $\mathbf{g}_{Q^u} \in \mathbb{R}^{m \times d}$. Let $i$ denote the item index for the non-zero update. The SecAgg can be performed using the following steps:

**Step 1: Encode non-zero elements with $d$ point functions, $\mathbf{g}_{Q^u_{ik}} \to f_{u,i,k}$ for $k \in [d]$.** User $u$ begins by encoding each element $\mathbf{g}_{Q^u_{ik}} \in \mathbb{R}$ with a point function $f_{u,i,k} : \mathcal{I} \to \mathbb{G}$. The function $f_{u,i,k}$ takes an item id $x \in \mathcal{I}$ as input and outputs $f_{u,i,k}(x) = \mathbf{g}_{Q^u_{ik}} \in \mathbb{R}$ if $x = i$, and $0 \in \mathbb{R}$ elsewhere.

**Step 2: Generate keys for the point function.** User $u$ performs FSS on $f_{u,i,k}$ and outputs $d$ pairs of keys, i.e.,

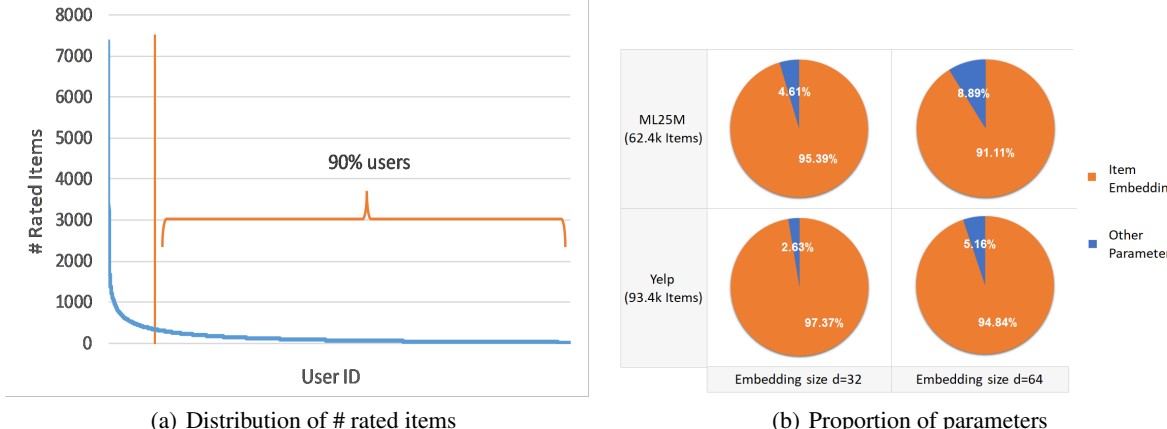

(a) Distribution of # rated items        (b) Proportion of parameters

*Figure 2.* (a) The long-tailed distribution of the number of rated items in ML10M. The x-axis is the user id sorted by their activeness, and the y-axis represents the number of rated items for the user. For ML10M dataset with 27,278 items, nearly 90% users have rated only up to 300 items. (b) The proportion of item embeddings and other parameters within a three-layer deep factorization machine (DeepFM) for ML25M and Yelp, under two different embedding sizes $d = 32$ and $d = 64$.

$(agK_{u,k}^0, agK_{u,k}^1) = \text{FSS.Gen}(1^\lambda, f_{u,i,k})$, which are then sent to their corresponding servers.

**Step 3: Aggregate secret shares from users.** On receiving $agK_{u,k}^s$ from all participating users, each server $s \in \{0,1\}$ computes their secret shares of the aggregated matrix as follows:

$$\mathbf{v}_{Q_{jk}}^s = \sum_u \text{FSS.Eval}(agK_{u,k}^s, j), \ \forall j \in \mathcal{I}, \ k \in [d]. \quad (5)$$

**Step 4: Reconstruct gradient aggregation.** The two servers can collaborate to reconstruct the plaintext aggregation matrix. To be specific, server 1 sends the aggregated secret shares to server 0, and the plaintext aggregation can be recover by:

$$\mathbf{g}_Q = \mathbf{v}_Q^0 + \mathbf{v}_Q^1. \quad (6)$$

Below we extends the method to cases where $m'_u > 1$.

**SecAgg for $m'_u > 1$:** In step 1, user $u$ generates $m'_u d$ point functions $f_{u,i,k} : \mathcal{I} \to \mathbb{G}$ for $i \in [m'_u]$. Let $\text{idx}(x)$ denote the global index of the $i$-th rated item. Accordingly, $f_{u,i,k}$ takes a item id $x \in \mathcal{I}$ as input and outputs $f_{u,i,k}(x) = \mathbf{g}_{Q_{\text{idx}(i),k}^u} \in \mathbb{R}$ if $x = \text{idx}(i)$, and $0 \in \mathbb{R}$ elsewhere. In step 2, user $u$ produces $m'_u d$ pairs of secret keys $(agK_{u,i,k}^0, agK_{u,i,k}^1)$ for $i \in [m'_u], k \in [d]$. In step 3, each server $s \in \{0,1\}$ computes their secret shares of the aggregated matrix as follows:

$$\mathbf{v}_{Q_{jk}}^s = \sum_u \sum_{i \in [m'_u]} \text{FSS.Eval}(agK_{u,i,k}^s, j), \forall j \in \mathcal{I}, k \in [d]. \quad (7)$$

### 4.2.3. ANALYSIS OF INITIAL CONSTRUCTION

We briefly analyze the complexity and security for the above construction as follows.

**Commmunication cost:** During privacy-preserving embedding retrieval, only $m'_u$ keys and embeddings are exchanged between user $u$ and server, rather than the entire embedding matrix. For update aggregation, user $u$ uploads only $m'_u d$ keys to each server instead of the whole sparse matrix. Therefore, the communication cost is approximately $O(m'_u d \cdot |\text{Key}|)$ for upload transmission and $O(m'_u d)$ for download transmission, where $|\text{Key}|$ denotes the key size.

**Computation cost:** The user computation cost primarily stems from the generation of FSS keys. In total, user $u$ generates $m'_u(d+1)$ keys, resulting in a computation overhead of $O(m'_u d \cdot \text{FGen})$, where FGen represents the cost of generating a single key.

**Security:** FSS security ensures that user updates and interactions remain hidden from the server. In privacy-preserving embedding retrieval, each server is ignorant of the targeted item indices. During update aggregation, servers learn no information about the rated item index $i$ and its gradient $\mathbf{g}_{Q_i^u}$. To further hide the number of rated items $m'_u$ from servers, we can pre-specify a unified update size $m'$, and accordingly pad or truncate the target indices as well as updated vectors to contain $m'$ items (see Appendix C), thus keeping the entire sparse update matrix and target item set hidden from the server.

### 4.3. Optimization of SecEmb

We identify a crucial property of the FSS key—for party $b \in \{0,1\}$, the key, denoted as $s_b^0||t_b^0||CW^1||\cdots||CW^{n+1}$

(where $n$ is the bit length of the input), consists of two components: (1) $s_b^0||t_b^0||CW^1||\cdots||CW^n$, the seed and correction words used to determine whether the input index corresponds to the non-zero element, and (2) $CW^{n+1}$, the correction word used to convert the final seed into group elements in the abelian group $\mathbb{G}$ (see Figure 3). The former part can be identical for point functions with the same non-zero index, presenting an opportunity to eliminate redundancy in FSS keys. Leveraging this insight, we propose two optimizations of SecEmb to further reduce user payload. Algorithm 3 outlines the improved SecEmb.

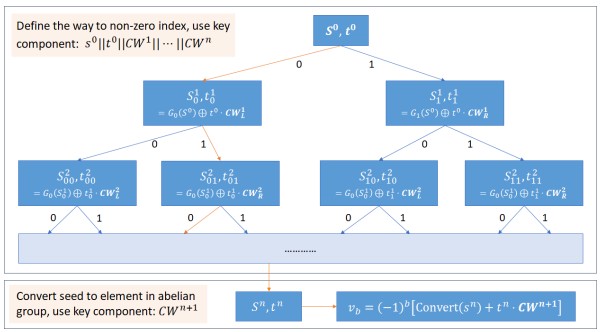

*Figure 3.* Evaluation of FSS scheme for party $b \in \{0, 1\}$. For simplification, we use $s_b^0$ ($t_b^0$) and $s^0$ ($t^0$) interchangeably.

### 4.3.1. EFFICIENT ROW-WISE ENCODING

The gradient update of item embeddings forms a *row-wise sparse matrix*, with a few rows containing non-zero values. The initial method generate FSS keys separately for each element, introducing redundancy for updates from the same item. A more efficient approach is to encode each non-zero row as a point function, requiring only a single key pair per embedding update (see Figure 4).

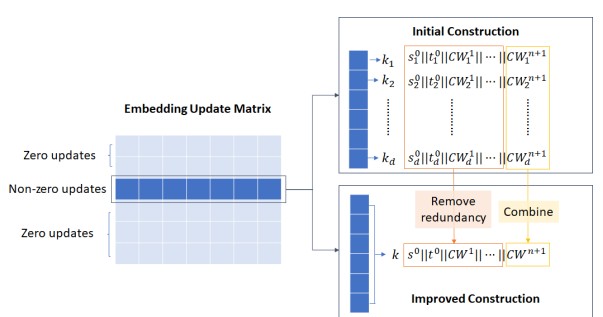

*Figure 4.* Row-wise encoding of item embedding gradient.

Specifically, user $u$ encodes the embedding gradients for targeted items into $m'$ point functions $f_{u,i} : \mathcal{I} \to \mathbb{G}$ for $i \in [m']$. The function $f_{u,i}$ takes an item id $x \in \mathcal{I}$ as input and outputs $f_{u,i}(x) = \mathbf{g}_{Q_i^u} \in \mathbb{R}^d$ if $x = \text{idx}(i)$, and $\mathbf{0} \in \mathbb{R}^d$ elsewhere. Then $m'$ pairs of FSS keys are generated correspondingly, i.e., $(agK_{u,i}^0, agK_{u,i}^1) = \text{FSS.Gen}(1^\lambda, f_{u,i})$ for $i \in [m']$. Each server $s \in \{0, 1\}$ computes their secret

shares of the aggregated matrix as follows:

$$\mathbf{v}_{Q_j}^s = \sum_u \sum_{i \in [m']} \text{FSS.Eval}(agK_{u,i}^s, j), \quad \forall j \in \mathcal{I}. \quad (8)$$

After the optimization, each user generates and uploads only $m'$ keys instead of $m'd$ keys for update aggregation.

### 4.3.2. SHARING PATH FROM RETRIEVAL STAGE

For each user, *the indices of relevant embeddings remain the same across both stages.* If user $u$ generates an embedding retrieval key $reK_{u,i}^s$ and an update aggregation key $agK_{u,i}^s$ for item $i$, the former components of both keys can be identical.

Consequently, in the update aggregation stage, each user can generate and transmit $CW^{n+1}$ instead of the whole FSS key, leading to much lower computation and communication cost. Furthermore, server $s \in \{0, 1\}$ can reuse the binary tree path identified in private embedding retrieval stage to compute $\text{FSS.Eval}(agK_{u,i}^s, j)$ for $u \in \mathcal{U}, j \in \mathcal{I}, i \in [m']$.

The optimization reduces the key size in the update aggregation stage from $n + 2$ seeds and corrections words to a single correction word. Additionally, each user eliminates $n$ AES operations for pseudorandom generation when secret sharing the embedding update.

### 4.4. Complexity and Security Analysis

#### 4.4.1. COMPLEXITY ANALYSIS

Denote $\lambda$ as the security parameter of FSS scheme, and $b$ as the number of bits required to represent a single numerical value. The variables $d$ and $\theta$ refer to the embedding size and the parameters other than item embeddings. The keys sizes for the private embedding retrieval and secure aggregation stages are $(\lambda + 2) \log m$ and $bd$, respectively (Boyle et al., 2016). The computation cost to generate an FSS key in private embedding retrieval stage is $O(\log m \cdot \text{AES})$, and the cost to derive partial key in upload aggregation is negligible compared with the AES operations. Considering $m'$ functions and $|\theta|$ dense updates, we have upload communication complexity of $O\left(m'(bd + \lambda \log m) + |\theta|b\right)$, download complexity of $O\left(m'bd + |\theta|b\right)$, and computation cost of $O\left(m' \log m \cdot \text{AES} + |\theta|\right)$.

Table 1 compares the user-side cost between SecEmb and secure FedRec (Xiong et al., 2020). Secure FedRec utilizes full model download and adopts the most efficient SecAgg for update aggregation (see Table 6). The communication cost of SecEmb scales linearly with $m'$ and logarithmically with $m$. SecEmb outperforms secure FedRec in upload communication cost as long as $m' < mbd/\left((\lambda + 2) \log m + b(d + 1)\right)$, and in download cost as long as $m' < m/2$. In Appendix K.5 we demonstrate that

these inequalities usually hold for RecSys with sparse update. The computation complexity of SecEmb primarily arises from AES operations, which can be mitigated when $m'$ is sufficiently small compared with $m$.

*Table 1.* User computation and communication cost of SecEmb and secure FedRec. CommunDown and CommunUp refer to download and upload communication cost, respectively.

| | SecEmb | Secure FedRec |
|---|---|---|
| CommunDown | $O\left(m'bd + |\theta|b\right)$ | $O\left(mbd + |\theta|b\right)$ |
| CommunUp | $O\left(m'(\lambda \log m + bd) + |\theta|b\right)$ | $O\left(mbd + |\theta|b\right)$ |
| Computation | $O\left(m' \log m \cdot \text{AES} + |\theta|\right)$ | $O\left(md + |\theta|\right)$ |

### 4.4.2. SECURITY ANALYSIS

The FSS security property ensures that the two non-colluding servers learn no more information than just the aggregated gradients. The FSS keys hide the rated item index as well as the values of updated gradients from each server. Under a pre-determined $m'$, servers are ignorant about the number of rated item for each user. Consequently, no information about the individual updates is revealed to the servers except the aggregation. Below is the formal security formulation for the update aggregation stage; a comprehensive security analysis is provided in Appendix F.

For $\mathcal{C} \subset \mathcal{U} \cup \{b\}$ ($b \in \{0, 1\}$), let $\text{Real}_{\lambda, agg}^{\mathcal{C}}$ be the joint view of colluding parties $\mathcal{C}$ in experiments executing secure update aggregation in SecEmb. Denote $\{(\mathbf{g}_{Q^u}, \theta_u)\}_{u \in \mathcal{U}}$ as the set of gradients from all users. Then we have the security of the aggregation protocol as follows.

**Theorem 4.1** (Security of secure update aggregation). *There exists a PPT simulator $Sim_{\lambda, agg}^{\mathcal{C}}$, such that for all user input $\{(\mathbf{g}_{Q^u}, \theta_u)\}_{u \in \mathcal{U}}$ and $\mathcal{C} \subset \mathcal{U} \cup \{b\}$ ($b \in \{0, 1\}$), the output of $Sim_{\lambda, agg}^{\mathcal{C}}$ and $\text{Real}_{\lambda, agg}^{\mathcal{C}}$ are computationally indistinguishable:*

$$\text{Real}_{\lambda, agg}^{\mathcal{C}}(1^{\lambda}, \{(\mathbf{g}_{Q^u}, \theta_u)\}_{u \in \mathcal{U} \setminus \mathcal{C}})$$
$$= Sim_{\lambda, agg}^{\mathcal{C}}(1^{\lambda}, (m', \mathbb{G}_1, ..., \mathbb{G}_{m'}, \mathbf{g}_{\theta}, \mathbf{g}_Q)), \quad (9)$$

*where $\mathbf{g}_Q$ and $\mathbf{g}_{\theta}$ denote the aggregated gradients for item embedding and the remaining parameters, respectively.*

It is important to note that our algorithm can be integrated with differential privacy (DP) to achieve stronger privacy protection. In particular, each server can independently add calibrated noises to the aggregated secret shares matrix, so that recovered aggregation matrix adheres to $(\epsilon, \delta)$-DP (Dwork, 2006; Cormode et al., 2018) (see Appendix G).

## 5. Experiment Evaluation

### 5.1. Experiment Setting

We evaluate our SecEmb on five public datasets: Movie-Lens 100K (ML100K), MovieLens 1M (ML1M), Movie-

*Table 2.* Upload communication cost (in MB) per user for SecEmb and Secure FedRec (SecFedRec) in one iteration. Reduction ratio is computed as the communication overhead of SecFedRec by that of SecEmb.

| | ML100K | ML1M | ML10M | ML25M | Yelp |
|---|---|---|---|---|---|
| Item Size $m$ | 1,682 | 3,883 | 10,681 | 62,423 | 93,386 |
| | | | MF | | |
| SecFedRec | 0.86 | 1.99 | 5.47 | 31.96 | 47.81 |
| SecEmb | 0.17 | 0.27 | 0.28 | 0.51 | 0.52 |
| Red. Ratio | 4.99 | 7.35 | 19.22 | 62.07 | 91.22 |
| | | | NCF | | |
| SecFedRec | 0.44 | 1.00 | 4.11 | 23.98 | 29.89 |
| SecEmb | 0.13 | 0.20 | 0.26 | 0.46 | 0.44 |
| Red. Ratio | 3.44 | 5.02 | 15.93 | 51.80 | 68.46 |
| | | | FM | | |
| SecFedRec | 0.91 | 2.01 | 5.48 | 31.97 | 47.82 |
| SecEmb | 0.18 | 0.28 | 0.29 | 0.53 | 0.53 |
| Red. Ratio | 5.05 | 7.22 | 18.76 | 60.87 | 91.05 |
| | | | DeepFM | | |
| SecFedRec | 14.95 | 8.84 | 8.63 | 35.13 | 50.45 |
| SecEmb | 14.22 | 7.10 | 3.45 | 3.68 | 3.16 |
| Red. Ratio | 1.05 | 1.24 | 2.50 | 9.54 | 15.98 |

Lens 10M (ML10M), MovieLens 25M (ML25M), and Yelp (Harper & Konstan, 2015; Yelp, 2015). For the Yelp dataset, we sample a portion of top users ranked in descending order by their number of rated items, and obtain a subset containing 10,000 users and 93,386 items. Table 7 summarizes the statistics for the datasets.

Our framework is tested with four latent factor-based recommender models: matrix factorization with biased term (MF) (Koren et al., 2009), neural collaborative filtering (NCF) (He et al., 2017), factorization machine (FM) (Rendle, 2010), and deep factorization machine (DeepFM) (Guo et al., 2017). Detailed hyperparameters for each model are provided in Appendix I. We provide a coarse-grained comparison between SecEmb and existing FL protocols in Appendix J.

### 5.2. Efficiency Analysis

#### 5.2.1. COMMUNICATION COST

To evaluate the communication efficiency of our framework, we conduct a comparative analysis of the communication payload between SecEmb and secure FedRec as presented in Table 2 and 12. We use the two-server ASS, which has the minimal communication overhead, to compute the upload cost for secure FedRec (see Table 6). A key finding is that SecEmb's communication overhead increases at a significantly slower rate with item size compared to secure FedRec, particularly for models characterized by a higher proportion of sparse updates.

For upload communication, our protocol reduces costs by approximately 4x to 90x for MF and FM, which have minimal dense updates, depending on the dataset item size. For

*Figure 5.* User computation time (in milliseconds) for secret shares generation during training phase.

NCF that includes a small share of dense updates, SecEmb achieves overhead reductions ranging from roughly 3.5x to 70x. For DeepFM, the reduction is less pronounced for the ML100K and ML1M datasets with item sizes lower than 4k, while the cost savings become more substantial as the item size exceeds 10k. A similar pattern is observed for download communication (see Appendix K.1).

We also compare our SecEmb with existing sparse aggregation protocols in Appendix K.3.

### 5.2.2. COMPUTATION COST

Figure 5 compares the user computation time to generate the secret shares for SecEmb and secure FedRec with the most efficient SecAgg protocol. SecEmb has an advantage over secure FedRec in terms of the computation overhead, since each user generates fewer shares when $m' \ll m$. Furthermore, the computation time for SecEmb scales more slowly with an increase in item size compared to secure FedRec. This results in a significantly higher reduction ratio for datasets encompassing a greater number of items, achieving nearly 70x for Yelp withs MF and FM.

### 5.3. Utility Analysis

We evaluate the utility of SecEmb against several lossy message compression methods for communication efficiency, including: (1) Singular value decomposition (SVD) (Nguyen et al., 2024), (2) Correlated Low-rank Structure (CoLR) (Nguyen et al., 2024), (3) 8-bit quantization (Bit8Quant) (Dettmers, 2015), and (4) Ternary Quantization (TernQuant) (Wen et al., 2017). The first two methods represent dimension reduction approaches, and the latter two employ gradient quantization method. For fair comparison, the lossy message compressions are applied only to item embeddings.

Table 3 presents the prediction accuracy and reduction ratio of communication cost. We focus on the upload communication as most baselines do not optimize download costs (see Table 11). It can be observed that: (1) While SecEmb

achieves similar reduction ratios on datasets with smaller item sizes, its communication benefits significantly surpass those of other compression methods on ML25M and Yelp, where the item size exceeds 60k. (2) The performance is degraded on an average by 1.29%, 1.74%, 1.95%, and 2.04% for Bit8Quant, TernQuant, SVD, and CoLR, respectively, suggesting that SecEmb offers non-negligible advantages over the lossy message compression mechanisms.

### 5.4. Application to Sequential Recommendation

We extend our framework to sequential recommendation tasks, which predict the next item a user will interact with based on their historical interactions. Specifically, we apply SecEmb to the item embedding layers of sequential recommendation models. We evaluate our approach using two sequence models, Caser (Tang & Wang, 2018) and SASRec (Kang & McAuley, 2018), on the ML1M and Amazon datasets. Experimental settings are detailed in Appendix K.6.

Our findings in Table 4 indicate that compared with SecEmb, applying existing message compression techniques results in average reductions of 0.5%, 2.7%, 2.6%, and 2.6% for Bit8Quant, TernQuant, SVD, and CoLR, respectively. Notably, for the Amazon dataset, which has a vast item set and exhibits high sparsity (density$< 0.002$‰), our SecEmb method achieves up to a 2500× reduction in upload communication cost.

### 5.5. Ablation Studies

To investigate the effectiveness of our optimizations, we compare SecEmb with two variants: (1) initial construction of SecEmb (SecEmb-Init), and (2) initial construction of SecEmb optimized by efficient row-wise encoding (SecEmb-RowEnc). Table 5 presents the user cost with MF model, and for full results refer to Section K.7. The initial construction of SecEmb suffers substantially higher communication overhead than the improved one, and its cost can be higher than that for secure FedRec on dataset with item size lower

*Table 3.* RMSE and Reduction Ratio (R.R.) for SecEmb and various message compression methods. The values for RMSE denote the mean $\pm$ standard deviation of four rounds of experiments. R.R. refers to the ratio of upload communication cost before and after the application of the compression mechanism.

| | | ML100K | | ML1M | | ML10M | | ML25M | | Yelp | |
|---|---|---|---|---|---|---|---|---|---|---|---|
| | | RMSE | R.R. | RMSE | R.R. | RMSE | R.R. | RMSE | R.R. | RMSE | R.R. |
| MF | Bit8quant | $0.948_{\pm0.004}$ | 4.00 | $0.914_{\pm0.000}$ | 4.00 | $0.872_{\pm0.001}$ | 4.00 | $0.870_{\pm0.000}$ | 4.00 | $1.050_{\pm0.001}$ | 4.00 |
| | Ternquant | $0.951_{\pm0.002}$ | 8.00 | $0.916_{\pm0.002}$ | 8.00 | $0.873_{\pm0.001}$ | 8.00 | $0.874_{\pm0.000}$ | 8.00 | $1.050_{\pm0.001}$ | 8.00 |
| | SVD | $0.952_{\pm0.006}$ | 5.22 | $0.917_{\pm0.001}$ | 6.39 | $0.872_{\pm0.001}$ | 16.15 | $0.873_{\pm0.000}$ | 16.23 | $1.050_{\pm0.001}$ | 16.24 |
| | CoLR | $0.951_{\pm0.001}$ | 5.42 | $0.916_{\pm0.001}$ | 6.50 | $0.872_{\pm0.001}$ | 16.25 | $0.873_{\pm0.000}$ | 16.25 | $\mathbf{1.049}_{\pm0.002}$ | 16.25 |
| | **SecEmb** | $\mathbf{0.944}_{\pm0.003}$ | 4.99 | $\mathbf{0.903}_{\pm0.002}$ | 7.35 | $\mathbf{0.868}_{\pm0.003}$ | 19.22 | $\mathbf{0.864}_{\pm0.002}$ | 62.07 | $1.050_{\pm0.001}$ | 91.22 |
| NCF | Bit8quant | $0.950_{\pm0.011}$ | 3.86 | $0.898_{\pm0.002}$ | 3.94 | $0.832_{\pm0.004}$ | 3.97 | $0.820_{\pm0.001}$ | 3.99 | $1.035_{\pm0.002}$ | 4.00 |
| | Ternquant | $0.951_{\pm0.005}$ | 7.37 | $0.901_{\pm0.007}$ | 7.71 | $0.833_{\pm0.001}$ | 7.84 | $0.825_{\pm0.002}$ | 7.97 | $1.036_{\pm0.004}$ | 7.98 |
| | SVD | $0.952_{\pm0.006}$ | 3.15 | $0.903_{\pm0.007}$ | 4.02 | $0.838_{\pm0.002}$ | 11.81 | $0.824_{\pm0.000}$ | 12.17 | $1.035_{\pm0.002}$ | 10.22 |
| | CoLR | $\mathbf{0.947}_{\pm0.005}$ | 3.21 | $0.912_{\pm0.005}$ | 4.06 | $0.856_{\pm0.000}$ | 11.87 | $0.839_{\pm0.001}$ | 12.18 | $\mathbf{1.032}_{\pm0.001}$ | 10.22 |
| | **SecEmb** | $0.949_{\pm0.014}$ | 3.44 | $\mathbf{0.897}_{\pm0.006}$ | 5.02 | $\mathbf{0.819}_{\pm0.003}$ | 15.93 | $\mathbf{0.786}_{\pm0.008}$ | 51.80 | $1.035_{\pm0.001}$ | 68.46 |
| FM | Bit8quant | $0.945_{\pm0.001}$ | 3.41 | $0.912_{\pm0.000}$ | 3.86 | $\mathbf{0.845}_{\pm0.000}$ | 3.98 | $0.836_{\pm0.001}$ | 4.00 | $1.009_{\pm0.001}$ | 4.00 |
| | Ternquant | $0.948_{\pm0.002}$ | 5.70 | $0.913_{\pm0.001}$ | 7.37 | $0.856_{\pm0.000}$ | 7.90 | $0.850_{\pm0.000}$ | 7.98 | $1.010_{\pm0.002}$ | 7.99 |
| | SVD | $0.946_{\pm0.001}$ | 4.21 | $0.913_{\pm0.001}$ | 6.00 | $0.870_{\pm0.003}$ | 15.71 | $0.856_{\pm0.002}$ | 16.16 | $1.009_{\pm0.001}$ | 16.20 |
| | CoLR | $0.944_{\pm0.004}$ | 4.33 | $0.915_{\pm0.001}$ | 6.10 | $0.867_{\pm0.000}$ | 15.81 | $0.851_{\pm0.001}$ | 16.17 | $1.008_{\pm0.001}$ | 16.21 |
| | **SecEmb** | $\mathbf{0.937}_{\pm0.004}$ | 5.05 | $\mathbf{0.906}_{\pm0.000}$ | 7.22 | $0.848_{\pm0.002}$ | 18.76 | $\mathbf{0.789}_{\pm0.003}$ | 60.87 | $\mathbf{1.008}_{\pm0.003}$ | 91.05 |
| DeepFM | Bit8quant | $0.947_{\pm0.007}$ | 1.05 | $0.912_{\pm0.003}$ | 1.21 | $0.840_{\pm0.004}$ | 1.92 | $0.832_{\pm0.003}$ | 3.16 | $1.012_{\pm0.002}$ | 3.47 |
| | Ternquant | $0.949_{\pm0.004}$ | 1.06 | $0.914_{\pm0.002}$ | 1.25 | $0.847_{\pm0.000}$ | 2.26 | $0.842_{\pm0.001}$ | 4.94 | $1.018_{\pm0.001}$ | 5.88 |
| | SVD | $0.954_{\pm0.005}$ | 1.05 | $0.913_{\pm0.001}$ | 1.24 | $0.853_{\pm0.002}$ | 2.48 | $0.847_{\pm0.000}$ | 6.90 | $1.014_{\pm0.001}$ | 9.09 |
| | CoLR | $0.952_{\pm0.001}$ | 1.05 | $0.905_{\pm0.001}$ | 1.24 | $0.856_{\pm0.001}$ | 2.49 | $0.841_{\pm0.001}$ | 6.90 | $1.017_{\pm0.002}$ | 9.10 |
| | **SecEmb** | $\mathbf{0.939}_{\pm0.006}$ | 1.05 | $\mathbf{0.902}_{\pm0.001}$ | 1.24 | $\mathbf{0.821}_{\pm0.001}$ | 2.50 | $\mathbf{0.791}_{\pm0.001}$ | 9.54 | $\mathbf{1.011}_{\pm0.002}$ | 15.98 |

*Table 4.* RMSE and Reduction Ratio (R.R.) for SecEmb and various message compression methods on Sequential Recommender System. R.R. refers to the ratio of upload communication cost before and after the application of the compression mechanism.

| | | ML1M (3,883 items) | | | Amazon (9,267,503 items) | | |
|---|---|---|---|---|---|---|---|
| | | HR@10 | NDCG@10 | R.R. | HR@10 | NDCG@10 | R.R. |
| Caser | Bit8quant | 0.473 | 0.267 | 3.51 | 0.628 | 0.481 | 4 |
| | Ternquant | 0.468 | 0.261 | 6.03 | 0.623 | 0.478 | 8 |
| | SVD | 0.471 | 0.266 | 9.71 | 0.627 | 0.483 | 38 |
| | CoLR | 0.465 | 0.263 | **9.78** | 0.625 | 0.480 | 38 |
| | **SecEmb** | **0.475** | **0.270** | 4.79 | **0.629** | **0.483** | **2549** |
| SASRec | Bit8quant | 0.468 | 0.266 | 2.55 | 0.636 | 0.485 | 4 |
| | Ternquant | 0.443 | 0.248 | 3.44 | 0.635 | 0.483 | 8 |
| | SVD | 0.444 | 0.249 | 3.00 | 0.621 | 0.472 | 12 |
| | CoLR | 0.464 | 0.262 | 3.02 | 0.600 | 0.463 | 12 |
| | **SecEmb** | **0.472** | **0.268** | **3.86** | **0.639** | **0.485** | **2342** |

than 11k. Additionally, sharing the binary path between the two modules reduces the communication cost by around 36%. Similarly, the two optimizations substantially improve user computation costs.

*Table 5.* Upload communication cost and computation cost for secret generation per user for SecEmb and its variants with MF.

| | ML100K | ML1M | ML10M | ML25M | Yelp |
|---|---|---|---|---|---|
| | Upload communication cost (in MB) | | | | |
| SecEmb-Init | 4.56 | 7.60 | 8.51 | 16.83 | 17.43 |
| SecEmb-RowEnc | 0.28 | 0.43 | 0.45 | 0.78 | 0.79 |
| SecEmb | 0.17 | 0.27 | 0.28 | 0.51 | 0.52 |
| | Computation cost (in milliseconds) | | | | |
| SecEmb-Init | 5.34 | 15.35 | 22.56 | 28.53 | 26.43 |
| SecEmb-RowEnc | 0.37 | 0.90 | 0.91 | 1.34 | 1.47 |
| SecEmb | 0.31 | 0.47 | 0.47 | 0.72 | 0.74 |

## 6. Conclusion

This paper proposes SecEmb, a lossless privacy-preserving recommender system designed for resource-constrained devices with optimized payload on sparse embedding updates. SecEmb achieves succinct communication costs, i.e., cost independent of item size $m$ for downloads and logarithmic in $m$ for uploads, while preserving the secrecy of individual gradients. Additionally, it reduces device-side memory and computation overhead by processing only relevant item embeddings. The empirical evaluation demonstrates that: (1) SecEmb reduces communication costs by up to 90x and decreases user computation time by up to 70x compared with secure FedRec. (2) SecEmb offers non-negligible utility advantages compared with lossy message compression methods. Further discussions on our framework are provided in Appendix L.

## Acknowledgements

Ding was supported in part by a grant from the Israel Science Foundation (ISF Grant No. 1774/20), and by the European Union (ERC, SCALE,101162665). Views and opinions expressed are however those of the author(s) only and do not necessarily reflect those of the European Union or the European Research Council. Neither the European Union nor the granting authority can be held responsible for them.

## Impact Statement

This paper presents work aimed at advancing the field of privacy-preserving computation, particularly focusing on improving user privacy in federated recommender system through our SecEmb framework. Our proposed framework contributes to the technical evolution of federated learning and presents substantial benefits in data protection. We believe that the ethical impacts and social consequences align with established principles in responsible AI development.

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

## A. Complexity of Existing SecAgg Algorithms

In Table 6, we summarize the computation and communication complexity of existing SecAgg algorithms. It can be observed that a) two-server ASS represents the most efficient algorithm in terms of both computation and communication complexity, and b) the per-client communication cost depends linear in the model size $l$ for all protocols.

*Table 6.* Computation and communication complexity of existing SecAgg algorithms. SecAgg and SecAgg+ refer to the algorithm proposed by (Bonawitz et al., 2017) and (Bell et al., 2020) respectively. $n$ and $l$ denote client size and model size, respectively.

| | Server | | Client | | Rounds |
|---|---|---|---|---|---|
| | Computation | Communication | Computation | Communication | |
| SecAgg | $O(n^2 l)$ | $O(nl + n^2)$ | $O(nl + n^2)$ | $O(l + n)$ | 4 |
| SecAgg+ | $O(nl \log n + n \log^2 n)$ | $O(nl + n \log n)$ | $O(l \log n + \log^2 n)$ | $O(l + \log n)$ | 3 |
| FastSecAgg | $O(l \log n)$ | $O(nl + n^2)$ | $O(l \log n)$ | $O(l + n)$ | 3 |
| LightSecAgg | $O(nl \log^2 n)$ | $O(nl)$ | $O(nl \log^2 n)$ | $O(nl)$ | 2 |
| SAFELearn | $O(nl)$ | $O(nl)$ | $O(l)$ | $O(l)$ | 2 |
| Two-server ASS | $O(nl)$ | $O(nl)$ | $O(l)$ | $O(l)$ | 1 |

## B. Preliminaries

### B.1. Additive Secret Sharing

Additive secret sharing (ASS) (Cramer et al., 2015) divides a secret $x \in \mathbb{F}_p$ from a finite field into $n$ shares, such that $\sum_{i=1}^{n} x_i \pmod{p} = x$. Consequently, any $n - 1$ shares reveal nothing about the secret $s$. Furthermore, given two secret shares $[\![x]\!] = (x_1, ..., x_n)$ and $[\![y]\!] = (y_1, ..., y_n)$ from $\mathbb{F}_p$, it holds that $[\![x + y]\!] = (x_1 + y_1, ..., x_n + y_n)$.

### B.2. Function Secret Sharing

In this section we formally define the correctness and security properties of FSS scheme.

**Definition B.1** (FSS Correctness and Security). Let $\text{FSS} = (\text{FSS.Gen}, \text{FSS.Eval})$ be a FSS scheme for a function class $\mathcal{F}$, satisfying the following properties:

- **Correctness:** For every $x$ in the domain of $f$, it holds that:

$$\Pr \left( \sum_{i=1}^{2} \text{FSS.Eval}(k_i, x) = f(x) \in \mathbb{F} : (k_1, k_2) \leftarrow \text{FSS.Gen}(1^\lambda, f) \right) = 1. \tag{10}$$

- **Security:** For any party $s \in \{1, 2\}$, there exists a PPT algorithm Sim (simulator), such that for every function $f \in \mathcal{F}$, the outputs of the following experiments REAL and IDEAL are computationally indistinguishable:

    - $\text{REAL}(1^\lambda, f) = \{k_s : (k_1, k_2) \leftarrow \text{FSS.Gen}(1^\lambda, f)\}$
    - $\text{IDEAL}(1^\lambda, f, \mathcal{F}) = \{k_s \leftarrow \text{Sim}(1^\lambda, \mathcal{F})\}$

## C. Standardization of Uploaded Item Size

To conceal $m'_u$ from the server, a uniform $m'$ can be applied to all users. An optimal $m'$ should be substantially smaller than $m$ to reduce communication overhead, yet not excessively small to encompass the rated items of a majority of users. To determine a suitable value of $m'$, the server can compute the average number of rated items from all users via a SecAgg protocol and select $m'$ as follows:

$$m' = \alpha \cdot \frac{1}{n} \cdot \sum_u m'_u, \tag{11}$$

where $\alpha$ is a pre-specified multiplier on the average. Note that the SecAgg operation on the number of rated items is cheap, incurring $O(1)$ communication and computation overheads per user.

Given the unified $m'$, each user can: (1) convert their target item indices set into size of $m'$ according to Algorithm 1, and (2) standardize their non-zero updates for item embedding to be a $m' \times d$ matrix according to Algorithm 2.

---

**Algorithm 1** PadOrTruncIdx

---

    **Input:** $m'$ and $\mathcal{I}_u = \{i_1, ..., i_{m'_u}\}$.
    **Output:** $\mathcal{I}'_u = \{i_1, ..., i_{m'}\}$.
    **if** $|\mathcal{I}_u| < m'$ **then**
        Randomly sample $m' - |\mathcal{I}_u|$ elements from $\mathcal{I} \setminus \mathcal{I}_u$, and insert them into $\mathcal{I}_u$ to form $\mathcal{I}'_u$.
    **else if** $m'_u > m'$ **then**
        Randomly sample $m'$ elements from $\mathcal{I}_u$ to form $\mathcal{I}'_u$
    **else**
        Let $\mathcal{I}'_u = \mathcal{I}_u$
    **end if**
    **return** $\mathcal{I}'_u$

---

**Algorithm 2** PadOrTruncEmb

---

    **Input:** $m'$ and $\mathbf{g}_{Q_u} \in \mathbb{R}^{m'_u \times d}$.
    **Output:** $\mathbf{g}'_{Q_u} \in \mathbb{R}^{m' \times d}$.
    **if** $m'_u < m'$ **then**
        Create padding matrix of zero elements $\mathbf{0} \in \mathbb{R}^{(m' - m'_u) \times d}$
        Concatenate $\mathbf{g}_{Q_u}$ and $\mathbf{0}$ to form $\mathbf{g}'_{Q_u} \in \mathbb{R}^{m' \times d}$
    **else if** $m'_u > m'$ **then**
        Randomly sample $m'$ rows from $\mathbf{g}_{Q_u}$ to form $\mathbf{g}'_{Q_u} \in \mathbb{R}^{m' \times d}$
    **else**
        Let $\mathbf{g}'_{Q_u} = \mathbf{g}_{Q_u}$
    **end if**
    **return** $\mathbf{g}'_{Q_u}$

---

## D. Secure Aggregation on Dense Update

We employ additive secret sharing for SecAgg on the dense update $\mathbf{g}_\theta$. In particular, user $u$ generates a pair of additive secret shares for the gradients $[\![\mathbf{g}_\theta]\!] = (\mathbf{v}_\theta^1, \mathbf{v}_\theta^2)$, and sends the secret shares to the corresponding servers. Each server $s$ aggregates the secret shares from all participating users:

$$\mathbf{v}_\theta^s = \sum_u \mathbf{v}_{\theta_u}^s \tag{12}$$

Same as step 4 in Section 4.2.2, the two servers can subsequently collaborate to reconstruct the plaintext aggregated update.

## E. Algorithm of SecEmb

Algorithm 3 outlines the final version of SecEmb, which consists of two modules: (1) privacy-preserving embedding retrieval, and (2) secure update aggregation.

## F. Formal Security Analysis

For security, we require that the two servers should be non-colluding. However, no restriction is placed on the collusion between one server and the clients. User privacy is guaranteed as long as at least one server is honest, even if the other colludes with any number of clients.

### F.1. Security Analysis for Private Embedding Retrieval

Let $\mathcal{C} \subset \mathcal{U} \cup \{b\}$ ($b \in \{0, 1\}$) denote the union of one server and any subset of the users. Given a security parameter $\lambda$, let $\text{Real}_{\lambda, ret}^{\mathcal{C}}$ be the combined view of colluding parties $\mathcal{C}$ in experiments executing privacy-preserving embedding retrieval in SecEmb. Denote $\mathcal{I}_\mathcal{U}$ as the target indices of all users. We show that the joint view of each non-colluding server and the clients can be simulated given the allowable leakage. In other words, the collusion between one server and any number of clients reveals no information about the honest clients.

**Algorithm 3** SecEmb

**Server** $s \in \{0,1\}$:
**Initialize** public parameters $\Theta_s$.
**for** $t \in [1,T]$ **do**
  **Privacy-preserving embedding retrieval:**
  - Receive FSS keys $\{reK_{u,i}^s\}_{i \in [m']}$ from users $u \in \mathcal{A}_t$.
  - Compute secret shares of item embedding via equation 3, and store immediate values from the evaluation
  $(t_{u,i,j}, s_{u,i,j}) = \text{FSS.PathEval}(s, reK_{u,i}^s, j)$.
  - Send secret shares $\{\mathbf{v}_{u,i}^s\}_{i \in [m']}$ to users $u \in \mathcal{A}_t$.
  **Secure update aggregation:**
  - Receive partial FSS keys $\{CW_{u,i}\}_{i \in [m']}$ and secret shares for dense update $\mathbf{v}_{\theta_u}^s$ from users $u \in \mathcal{A}_t$.
  - Compute the secret shares of the aggregated sparse update via:
    $\mathbf{v}_{Q_j}^s = \sum_u \sum_{i \in [m']} \text{FSS.ConvertEval}(s, t_{u,i,j}, s_{u,i,j}, CW_{u,i})$ for $j \in \mathcal{I}$.
  - Aggregate the secret shares of the dense update via equation 12.
  - **if** $s = 0$ **then**
    - Receive the aggregated secret shares $(\mathbf{v}_Q^1, \mathbf{v}_\theta^1)$ from server 1.
    - Recover the gradients for public parameters $\mathbf{g}_Q, \mathbf{g}_\theta$.
    - Update public parameters $\Theta_s = (Q, \theta)$ with the gradients.
    - Synchronize public parameters with server 1.
  - **else**
    - Send the aggregated secret shares $(\mathbf{v}_Q^1, \mathbf{v}_\theta^1)$ to server 0.
    - Receive updated public parameters from server 0.
  - **end if**
**end for**

**User** $u \in \mathcal{U}$:
**for** $t \in [1,T]$ **do**
  **if** $u \in \mathcal{A}_t$ **then**
    **Privacy-preserving embedding retrieval:**
    - Standardize the size of target indices, i.e., $\mathcal{I}_u' = \text{PadOrTruncIdx}(m', \mathcal{I}_u)$.
    - Encode each index in $\mathcal{I}_u'$ with a point function, obtaining $\{f_{u,i}\}_{i \in [m']}$.
    - Generate FSS keys $(reK_{u,i}^0, reK_{u,i}^1) = \text{FSS.Gen}(1^\lambda, f_{u,i})$ for $i \in [m']$, and store the intermediate values from the generation $(t_{u,i}^1, s_{u,i}^0, s_{u,i}^1) = \text{FSS.PathGen}(1^\lambda, \text{idx}(i))$ for $i \in [m']$.
    - Send $\{rek_{u,i}^s\}_{i \in [m']}$ to server $s \in \{0,1\}$.
    - Receive $\{\mathbf{v}_{u,i}^s\}_{i \in [m']}$ from server $s \in \{0,1\}$, discard irrelevant embeddings, and reconstruct targeted embedding $Q^u$ using equation 4.
    **Secure update aggregation:**
    - Calculate gradients locally and update private parameters $\Theta_p$.
    - Construct additive secret shares $(\mathbf{v}_{\theta_u}^0, \mathbf{v}_{\theta_u}^1)$ for dense gradient $\mathbf{g}_{\theta_u}$.
    - Pad or truncate the sparse gradient into a $m' \times d$ matrix, $\mathbf{g}_{Q^u}' = \text{PadOrTruncEmb}(m', \mathbf{g}_{Q^u})$.
    - Generate partial FSS keys for the sparse gradients $\{CW_{u,i}\}_{i \in [m']} = \text{FSS.ConvertGen}(1^\lambda, t_{u,i}^1, s_{u,i}^0, s_{u,i}^1, \mathbf{g}_{Q_i}')$ for $i \in [m']$.
    - Send $(\mathbf{v}_{\theta_u}^s, \{CW_{u,i}\}_{i \in [m']})$ to server $s \in \{0,1\}$.
  **end if**
**end for**

---

**Algorithm 4** FSS.ConvertEval

Let $\text{Convert}_\mathbb{G} : \{0,1\}^\lambda \to \mathbb{G}$ be a map converting a random $\lambda$-bit string to a pseudorandom group element of $\mathbb{G}$.

  **Input:** $b, t^{(n)}, s^{(n)}, CW^{(n+1)}$
  **Output:** $v \in \mathbb{G}$
  $v = (-1)^b \left[ \text{Convert}(s^{(n)}) + t^{(n)} \cdot CW^{(n+1)} \right]$
  **return** $v$

---

**Algorithm 5** FSS.ConvertGen

---

Let $\text{Convert}_{\mathbb{G}} : \{0,1\}^{\lambda} \rightarrow \mathbb{G}$ be a map converting a random $\lambda$-bit string to a pseudorandom group element of $\mathbb{G}$.

    **Input:** $1^{\lambda}, t_1^{(n)}, s_0^{(n)}, s_1^{(n)}, \beta \in \mathbb{G}$

    **Output:** $CW^{(n+1)} \in \mathbb{G}$

    $CW^{(n+1)} \leftarrow (-1)^{t_1^{(n)}} \left[ \beta - \text{Convert}(s_0^{(n)}) + \text{Convert}(s_1^{(n)}) \right]$

    **return** $CW^{(n+1)}$

---

**Algorithm 6** FSS.PathEval

---

Let $G : \{0,1\}^{\lambda} \rightarrow \{0,1\}^{2(\lambda+1)}$ a pseudorandom generator.

    **Input:** $b$, $k_b$ and $x$

    **Output:** $t^{(n)} \in \{0,1\}$ and $s^{(n)} \in \{0,1\}^{\lambda}$

    Parse $k_b = s^{(0)}||t^{(0)}||CW^{(1)}||\cdots||CW^{(n+1)}$

    **for** $i = 1$ to $n$ **do**

        Parse $CW^{(i)} = s_{CW}||t_{CW}^L||t_{CW}^R$

        $\tau^{(i)} \leftarrow G(s^{(i-1)}) \oplus \left( t^{(i-1)} \cdot [s_{CW}||t_{CW}^L||s_{CW}||t_{CW}^R] \right)$

        Parse $\tau^{(i)} = s^L||t^L||s^R||t^R \in \{0,1\}^{2(\lambda+1)}$

        **if** $x_i = 0$ **then**

            $s^{(i)} \leftarrow s^L, t^{(i)} \leftarrow t^L$

        **else**

            $s^{(i)} \leftarrow s^L, t^{(i)} \leftarrow t^L$

        **end if**

    **end for**

    **return** $(t^{(n)}, s^{(n)})$

---

**Algorithm 7** FSS.PathGen

---

Let $G : \{0,1\}^{\lambda} \rightarrow \{0,1\}^{2(\lambda+1)}$ a pseudorandom generator.

    **Input:** $1^{\lambda}$ and $\alpha$

    **Output:** $t_1^{(n)} \in \{0,1\}$, $s_0^{(n)} \in \{0,1\}^{\lambda}$, and $s_1^{(n)} \in \{0,1\}^{\lambda}$

    Let $\alpha = \alpha_1, ..., \alpha_n \in \{0,1\}^n$ be the bit decomposition of $\alpha$.

    Sample random $s_0^{(0)} \leftarrow \{0,1\}^{\lambda}$ and $s_1^{(0)} \leftarrow \{0,1\}^{\lambda}$.

    Sample random $t_0^{(0)} \leftarrow \{0,1\}$ and let $t_1^{(0)} \leftarrow t_0^{(0)} \oplus 1$.

    **for** $i = 1$ to $n$ **do**

        $s_0^L||t_0^L||s_0^R||t_0^R \leftarrow G(s_0^{(i-1)})$

        $s_1^L||t_1^L||s_1^R||t_1^R \leftarrow G(s_1^{(i-1)})$

        **if** $\alpha_i = 0$ **then**

            $\text{Keep} \leftarrow L, \text{Lose} \leftarrow R$

        **else**

            $\text{Keep} \leftarrow R, \text{Lose} \leftarrow L$

        **end if**

        $s_{CW} \leftarrow s_0^{\text{Lose}} \oplus s_1^{\text{Lose}}$

        $t_{CW}^L \leftarrow t_0^L \oplus t_1^L \oplus \alpha_i \oplus 1$

        $t_{CW}^R \leftarrow t_0^R \oplus t_1^R \oplus \alpha_i$

        $CW^{(i)} \leftarrow s_{CW}||t_{CW}^L||t_{CW}^R$

        $s_b^{(i)} \leftarrow s_b^{\text{Keep}} \oplus t_b^{(i-1)} \cdot s_{CW}$ for $b = 0, 1$

        $t_b^{(i)} \leftarrow t_b^{\text{Keep}} \oplus t_b^{(i-1)} \cdot t_{CW}^{\text{Keep}}$ for $b = 0, 1$

    **end for**

    **return** $(t_1^{(n)}, s_0^{(n)}, s_1^{(n)})$

---

**Theorem F.1** (Security of private embedding retrieval). *There exists a PPT simulator $Sim^{\mathcal{C}}_{\lambda,ret}$, such that for all user input $\mathcal{I}_{\mathcal{U}}$ and $\mathcal{C} \subset \mathcal{U} \cup \{b\}$ ($b \in \{0,1\}$), the output of $Sim^{\mathcal{C}}_{\lambda,ret}$ and $Real^{\mathcal{C}}_{\lambda,ret}$ are computationally indistinguishable:*

$$Real^{\mathcal{C}}_{\lambda,ret}(1^{\lambda}, \mathcal{I}_{\mathcal{U}\backslash\mathcal{C}}) = Sim^{\mathcal{C}}_{\lambda,ret}(1^{\lambda}, (m', \mathbb{G}_1, ..., \mathbb{G}_{m'})). \tag{13}$$

*Proof.* For each $j \in \{0, 1, ..., m'\}$, we consider a distribution $\text{Hyb}_j$ as follows:

- If $i \leq j$, each user $u \in \mathcal{U}\backslash\mathcal{C}$ construct $rek^b_{u,i}$ to server $b$ by: (1) sample $s^{(0)}_b \leftarrow \{0,1\}^{\lambda}$ at random, and $t^{(0)}_b = b$; (2) choose $CW^{(1)}, ..., CW^{(\lceil \log m \rceil)} \leftarrow \{0,1\}^{\lambda+2}$ at random; (3) sample $CW^{(\lceil \log m \rceil+1)} \leftarrow \mathbb{G}$ at random; (4) output $reK^s_{u,i} = s^{(0)}_b \|t^{(0)}_b\|CW^{(1)}\|\cdots\|CW^{(\lceil \log m+1\rceil)}$.
- If $i > j$, compute $reK^b_{u,i}$ to server $b$ honestly using the function secret sharing algorithm.
- The output of the experiment is $\{reK^b_{u,i}\}_{i\in[m']}$ to server $b \in \{0,1\}$.

The FSS security ensures that $\text{Hyb}_j$ and $\text{Hyb}_{j+1}$ are computationally indistinguishable for $j \in \{0, 1, ..., m'-1\}$. Note that $\text{Hyb}_0$ corresponds to the $Real^{\mathcal{C}}_{\lambda,ret}$ distribution in the execution of SecEmb, where as $\text{Hyb}_{m'}$ generates a completely random key.

Hence, we complete the proof. □

### F.2. Security Analysis for Secure Update Aggregation

*Proof.* For each $j \in \{0, 1, ..., m'\}$, we consider a distribution $\text{Hyb}_j$ as follows:

- Compute $\mathbf{v}^b_{\theta_u}$ honestly using ASS to server $b$.
- If $i \leq j$, sample $CW_{u,i} \leftarrow \mathbb{G}$ at random.
- If $i > j$, compute $CW_{u,i} \leftarrow \mathbb{G}$ honestly following steps in SecEmb.
- The output of the experiment is $\left(\mathbf{v}^b_{\theta_u}, \{CW_{u,i}\}_{i\in[m']}\right)$ to server $b \in \{0,1\}$.

The proof of Theorem 4.1 requires to demonstrate that $\text{Hyb}_j$ and $\text{Hyb}_{j+1}$ are computationally indistinguishable. To justify this argument, we consider a intermediate distribution $\text{Hyb}_{j\rightarrow j+1}$ between $\text{Hyb}_j$ and $\text{Hyb}_{j+1}$:

- Compute $\mathbf{v}^b_{\theta_u}$ honestly using ASS to server $b$.
- If $i < j$, sample $CW_{u,i} \leftarrow \mathbb{G}$ at random.
- If $i = j$, construct $CW_{u,i} = \text{HybCW}(1^{\lambda}, \text{idx}_u(i))$ via Algorithm 8, where $\text{idx}_u(i)$ denotes the global index of the $i$-th item for user $u$.
- If $i > j$, compute $CW_{u,i} \leftarrow \mathbb{G}$ honestly following steps in SecEmb.
- The output of the experiment is $\left(\mathbf{v}^b_{\theta_u}, \{CW_{u,i}\}_{i\in[m']}\right)$ to server $b \in \{0,1\}$.

The security of the pseudorandom generator $G$ ensures that $\text{Hyb}_j$ and $\text{Hyb}_{j\rightarrow j+1}$ are computationally indistinguishable (see Claim 3.7 in (Boyle et al., 2016)). The security of the pseudorandom Convert ensures that $\text{Hyb}_{j\rightarrow j+1}$ and $\text{Hyb}_{j+1}$ are computationally indistinguishable (see Claim 3.8 in (Boyle et al., 2016)).

Next, we consider the following distribution $\text{Hyb}_{m'+1}$:

- Sample $\mathbf{v}^b_{\theta_u} \leftarrow \mathbb{G}_{\theta}$ randomly to server $b$.
- Sample $CW_{u,i} \leftarrow \mathbb{G}$ at random for $i \in [m']$.
- The output of the experiment is $\left(\mathbf{v}^b_{\theta_u}, \{CW_{u,i}\}_{i\in[m']}\right)$ to server $b \in \{0,1\}$.

The properties of additive secret sharing guarantee that the distribution of $\text{Hyb}_{m'+1}$ is identical to $\text{Hyb}_{m'}$. Note that $\text{Hyb}_0$ corresponds to the $Real^{\mathcal{C}}_{\lambda,agg}$ distribution in the execution of SecEmb, where as $\text{Hyb}_{m'+1}$ outputs a set of random secrets and keys.

Hence, we complete the proof. □

---

**Algorithm 8** HybCW

---

Let $\text{Convert}_{\mathbb{G}} : \{0,1\}^\lambda \to \mathbb{G}$ be a map converting a random $\lambda$-bit string to a pseudorandom group element of $\mathbb{G}$.

   **Input:** $1^\lambda, \alpha$
   **Output:** $CW^{(n+1)}$
   Let $\alpha = \alpha_1, ..., \alpha_n \in \{0,1\}^n$ be the bit decomposition of $\alpha$.
   Sample $s_b^{(0)} \leftarrow \{0,1\}^\lambda$, and let $t_b^{(0)} = b$.
   **for** $i = 1$ to $\lceil \log m \rceil$ **do**
      Compute $s_b^L \| t_b^L \| s_b^R \| s_b^L = G(s_b^{(i-1)})$
      **if** $\alpha_i = 0$ **then**
         $\text{Keep} \leftarrow L, \text{Lose} \leftarrow R$
      **else**
         $\text{Keep} \leftarrow R, \text{Lose} \leftarrow L$
      **end if**
      Sample $CW^{(i)} \leftarrow \{0,1\}^{\lambda+2}$
      Parse $CW^{(i)} = s_{CW} \| t_{CW}^L \| t_{CW}^R$
      $s_b^{(i)} \leftarrow s_b^{\text{Keep}} \oplus t_b^{(i-1)} \cdot s_{CW}$
      $t_b^{(i)} \leftarrow t_b^{\text{Keep}} \oplus t_b^{(i-1)} \cdot t_{CW}^{\text{Keep}}$
   **end for**
   $CW^{(n+1)} \leftarrow (-1)^{t_1^{(n)}} \left[ \beta - \text{Convert}(s_0^{(n)}) + \text{Convert}(s_1^{(n)}) \right]$
   **return** $CW^{(n+1)}$

---

# G. Providing Differential Privacy

## G.1. Implementation of Differentially Private Protocol

During the update aggregation stage, SecEmb ensures that each server learns only the aggregated updates. Consequently, we only need to ensure that the aggregated updates-equivalent to the servers' view-satisfy differential privacy (DP) throughout the training process.

Our protocol is possible to achieve the record-level DP that obfuscates a single user-item interaction's contribution (Liu et al., 2022; Wei et al., 2020; Hua et al., 2015). To implement DP in the two-server setting, each user conducts per-sample clipping on their local gradients $\bar{\mathbf{g}} \leftarrow \mathbf{g} \cdot \max\{1, \Delta_2/\mathbf{g}\}$ (Abadi et al., 2016), and secret shares the clipped gradients for aggregation. On computing the secret shares of aggregated update $v_s$, each server $s \in \{0,1\}$ adds Gaussian noise independently $v_s' \leftarrow v_s + \mathcal{N}(0, \sigma^2)$. To achieve $(\epsilon, \delta)$-DP, the noise scale can be set as:

$$\sigma = \sqrt{\ln(1.25/\delta)} \cdot \Delta_2/\epsilon. \tag{14}$$

## G.2. Empirical Analysis

We deploy the SecEmb with $(\epsilon, \delta)$-DP guarantee on ML1M dataset under experiment settings described in Section 5.1, using batch sizes of 100 and 500. The privacy budget is analyzed with moments accountant (Abadi et al., 2016), which provides a tight bound over multiple iterations. Figure 6 shows that: (1) MF has the best trade-off between privacy and utility, where the uility loss is within 2% for $\epsilon \geq 2$. This suggests that models with fewer parameters better preserve utility under similar privacy guarantees. (2) Increasing the batch size improves the privacy-utility trade-off, as noise is amortized over more samples.

While DP is sufficient to provide formal privacy guarantee, combining DP with SecAgg reduces the overall magnitude of noise, thus offering significantly better privacy-utility trade-off than local DP. In Figure 7, we compare the utility between SecEmb and DP-FedRec with MF model. DP-FedRec operates without SecAgg, where each user independently adds Guassian noises to their uploaded gradients to satisfy $(\epsilon, \delta)$-local DP. Integrating SecAgg with DP improve the performance by over 57% under $\epsilon < 1.5$ on ML1M, and over 47% under $\epsilon < 13$ on Yelp.

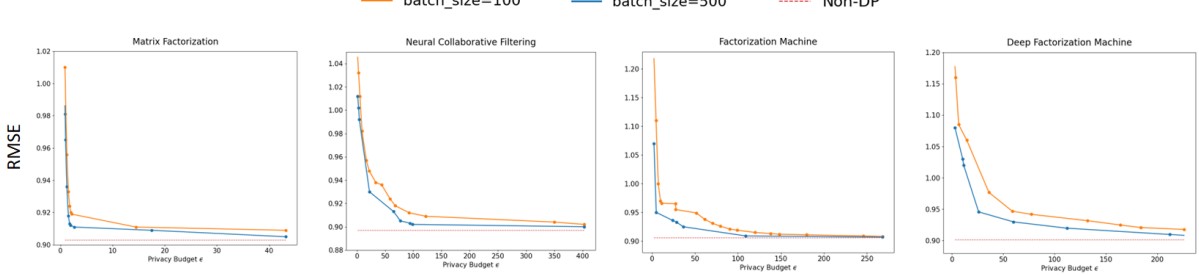

*Figure 6.* Performance on ML1M dataset under various differential privacy budget $\epsilon$.

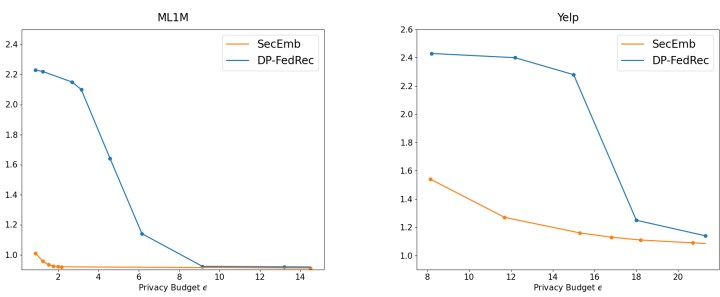

*Figure 7.* Performance of SecEmb and DP-FedRec under various privacy budget $\epsilon$ using MF.

## H. Dataset and Pre-processing

For each dataset, we encode the user and item features into binary vectors for model training. The features we select for binary encoding are given as follows:

- ML100K: movie genre, user gender, user age, and user occupation.

- ML1M: movie genre, user gender, user age, and user occupation.

- ML10M: movie genre.

- ML25M: movie genre.

- Yelp: restaurant state.

The statistics of the datasets are listed in Table 7.

*Table 7.* Statistics of the datasets. Yelp refers to the subset sampled from the whole dataset.

|  | # Users | # Items | # Ratings | # User Features | # Item Features | Density |
|---|---|---|---|---|---|---|
| ML100K | 943 | 1,682 | 100,000 | 84 | 19 | 6.30% |
| ML1M | 6,040 | 3,883 | 1,000,209 | 30 | 18 | 4.26% |
| ML10M | 69,878 | 10,681 | 10,000,054 | 0 | 20 | 1.34% |
| ML25M | 162,541 | 62,423 | 25,000,095 | 0 | 20 | 0.25% |
| Yelp | 10,000 | 93,386 | 1,007,956 | 0 | 16 | 0.11% |

## I. Hyperparameters of Recommender System

Each dataset is divided into 80% training and 20% testing data. For all cases, the recommender system is trained for 200 epochs. Each user represents an individual client and 100 clients are selected in each iteration. The parameters are updated

using Adaptive Moment Estimation (Adam) (Kingma, 2014) method. We use the combination of MSE and the regularization term as the loss function. The security parameter is set to $\lambda = 128$. Each experiment is run for four rounds and the average values are reported. Table 8 lists the specific hyperparameters for each dataset and model.

*Table 8.* Hyperparameters for federated training of recommender system.

|  |  | ML100K | ML1M | ML10M | ML25M | Yelp |
|---|---|---|---|---|---|---|
| MF | Embedding size | 64 | 64 | 64 | 64 | 64 |
|  | Learning rate | 0.025 | 0.025 | 0.01 | 0.01 | 0.01 |
|  | Regularization weight | 0.01 | 0.001 | 0.001 | 0.001 | 0.01 |
| NCF | Embedding size | 16 | 16 | 24 | 24 | 20 |
|  | Learning rate | 0.001 | 0.0001 | 0.001 | 0.001 | 0.001 |
|  | Regularization weight | 0.001 | 0 | 0 | 0 | 0 |
| FM | Embedding size | 64 | 64 | 64 | 64 | 64 |
|  | Learning rate | 0.025 | 0.025 | 0.005 | 0.005 | 0.01 |
|  | Regularization weight | 0.1 | 0.001 | 0 | 0 | 0.01 |
| DeepFM | Embedding size | 64 | 64 | 64 | 64 | 64 |
|  | Learning rate | 0.025 | 0.025 | 0.005 | 0.005 | 0.01 |
|  | Regularization weight | 0.1 | 0.001 | 0 | 0 | 0.001 |

For NCF, we fix the the architecture of the neural network layers to $2d \rightarrow d \rightarrow d/2$. For DeepFM, the neural network layers are fixed to $(l_x + l_y + 2)d \rightarrow 4d \rightarrow 2d$. We set the number of selected items $m'$ for ML100K, ML1M, ML10M, ML25M, and yelp as 200, 300, 300, 500, and 500, respectively. Table 9 presents the size of sparse and dense parameters, corresponding to the item embedding (including item bias term) and the remaining parameters.

*Table 9.* Size of dense and sparse parameters. # Sparse, # Non-zero Spr., and # Dense denote, respectively, the size of sparse update, size of non-zero elements in sparse update, and size of dense update.

|  |  | ML100K (1.7k Items) | ML1M (3.9k Items) | ML10M (10.7k Items) | ML25M (62.4k Items) | Yelp (93.4k Items) |
|---|---|---|---|---|---|---|
| MF | # Sparse | 109,330 | 252,395 | 694,265 | 4,057,495 | 6,070,090 |
|  | # Non-zero Spr. | 6,893 | 10,764 | 9,295 | 9,945 | 6,552 |
|  | # Dense | 0 | 0 | 0 | 0 | 0 |
| NCF | # Sparse | 55,506 | 128,139 | 523,369 | 3,058,727 | 3,828,826 |
|  | # Non-zero Spr. | 3,499 | 5,465 | 7,007 | 7,497 | 4,133 |
|  | # Dense | 688 | 688 | 1,512 | 1,512 | 1,060 |
| FM | # Sparse | 109,330 | 252,395 | 694,256 | 4,057,495 | 6,070,090 |
|  | # Non-zero Spr. | 6,893 | 10,764 | 9,295 | 9,945 | 6,552 |
|  | # Dense | 6,696 | 3,121 | 1,301 | 1,301 | 1,041 |
| DeepFM | # Sparse | 109,330 | 252,395 | 694,256 | 4,057,495 | 6,070,090 |
|  | # Non-zero Spr. | 6,893 | 10,764 | 9,295 | 9,945 | 6,552 |
|  | # Dense | 1,761,065 | 856,370 | 395,798 | 395,798 | 330,002 |

In FedRec using SVD and CoLR message compression techniques, the rank of the reduced matrix is specified in Table 10. A lower value of rank indicates higher level of reduction ratio.

*Table 10.* Rank for reduced update matrix in FedRec with SVD and CoLR compression techniques.

|  | ML100K | ML1M | ML10M | ML25M | Yelp |
|---|---|---|---|---|---|
| MF | 12 | 10 | 4 | 4 | 4 |
| NCF | 10 | 8 | 4 | 4 | 4 |
| FM | 12 | 10 | 4 | 4 | 4 |
| DeepFM | 12 | 10 | 4 | 4 | 4 |

## J. Comparison among FL Protocols

We compare our SecEmb with the following FL protocols:

- Secure FedRec that downloads full model and utilizes the most efficient SecAgg protocol.

- Two FL protocols with sparsity-aware aggregation on top-k sparsified gradients (Gupta et al., 2021; Aji & Heafield, 2017): Secure Aggregation with Mask Sparsification (SecAggMask) (Liu et al., 2023) and Top-k Sparse Secure Aggregation (TopkSecAgg) (Lu et al., 2023).

- Two quantization methods: 8-bit quantization (Bit8Quant) (Dettmers, 2015) and Ternary Quantization (TernQuant) (Wen et al., 2017).

- Two low-rank update methods: singular value decomposition (SVD) (Nguyen et al., 2024), and correlated Low-rank Structure (CoLR) (Nguyen et al., 2024).

Table 11 summarizes the comparison along four dimensions:

- **Secure FL training:** The training process should be secure, or compatible with SecAgg. In other words, the server learns no information about client updates, including non-zero indices and their values, except the aggregated model. SecAggMask and TopkSecAgg are not secure as they leak information about the non-zero indices. Kvsagg leaks more information to the server than SecEmb as it also exposes the exact number of clients who rated each item within a batch beside aggregated results. The quantization and low-rank methods are compatible with SecAgg protocols.

- **Reduced model download:** The client can download model in reduced size from the server, rather than the full model. Only CoLR and SecEmb allow to download the reduced model from the server.

- **Reduced gradient transmission:** The client can upload parameter gradients in reduced size to the server, rather than the entire gradients. The listed algorithms, except secure FedRec, compress the transmitted gradients for communication-efficient aggregation.

- **Operation on reduced model:** The client can store and operate on a reduced-size model locally for improved memory and computation efficiency. Unlike other algorithms, which assume users maintain a full model for local updates, SecEmb enables efficient operation without this requirement.

- **Lossless:** The training process should be lossless. Only secure FedRec and SecEmb are lossless in principle, and thus lossless in practice. Kvsagg has a minor failure rate, occasionally yielding inaccurate aggregation results, whereas SecEmb ensures error-free aggregation.

Among the FL protocols, only SecEmb simultaneously ensures efficiency, security, and utility for resource-constrained edge devices.

*Table 11.* Coarse-grained comparison among FL protocols.

|  | Secure FL training | Reduced model download | Reduced gradient transmission | Operation on reduced model | Lossless |
|---|---|---|---|---|---|
| Secure FedRec | ✓ | × | × | × | ✓ |
| SecAggMask | × | × | ✓ | × | × |
| TopkSecAgg | × | × | ✓ | × | × |
| Kvsagg | ✓× | × | ✓ | × | ✓× |
| Bit8Quant | ✓ | × | ✓ | × | × |
| TernQuant | ✓ | × | ✓ | × | × |
| SVD | ✓ | × | ✓ | × | × |
| CoLR | ✓ | ✓ | ✓ | × | × |
| **SecEmb** | ✓ | ✓ | ✓ | ✓ | ✓ |

# K. Additional Experiment Evaluation

## K.1. Download Communication Cost

The download communication cost is presented in Table 12. We compare our SecEmb against secure FedRec that downloads the full model for local update. For MF and NCF, our protocol reduces costs by approximately 4x to 90x, depending on dataset item size. For FM, SecEmb achieves overhead reductions ranging from roughly 10x to 116x. In DeepFM, cost savings are modest (within 2x) for ML100K and ML1M (item size less than 4k) but become more significant as item size increases, reaching around 20x for the Yelp dataset.

*Table 12.* Download communication cost (in MB) per user for SecEmb and Secure FedRec in one iteration. Reduction ratio is computed as the communication overhead of Secure FedRec by that of SecEmb.

|  |  | ML100K (1.7k Items) | ML1M (3.9k Items) | ML10M (10.7k Items) | ML25M (62.4k Items) | Yelp (93.4k Items) |
|---|---|---|---|---|---|---|
| MF | Secure FedRec | 0.43 | 0.99 | 2.73 | 15.98 | 23.91 |
|  | SecEmb | 0.10 | 0.15 | 0.15 | 0.26 | 0.26 |
|  | Reduction Ratio | 4.21 | 6.47 | 17.80 | 62.42 | 93.39 |
| NCF | Secure FedRec | 0.22 | 0.50 | 2.06 | 11.99 | 14.95 |
|  | SecEmb | 0.05 | 0.08 | 0.12 | 0.20 | 0.44 |
|  | Reduction Ratio | 4.04 | 6.28 | 16.96 | 60.55 | 91.00 |
| FM | Secure FedRec | 1.12 | 1.74 | 3.59 | 20.97 | 29.88 |
|  | SecEmb | 0.11 | 0.16 | 0.16 | 0.26 | 0.26 |
|  | Reduction Ratio | 10.50 | 11.00 | 22.76 | 80.32 | 116.50 |
| DeepFM | Secure FedRec | 8.14 | 5.15 | 5.17 | 22.55 | 31.20 |
|  | SecEmb | 7.12 | 3.57 | 1.74 | 1.84 | 1.57 |
|  | Reduction Ratio | 1.14 | 1.44 | 2.98 | 12.26 | 19.84 |

## K.2. Training Memory and Storage Analysis

In Figure 8 we present the training memory and storage cost for two cases: (1) SecEmb where each user utilizes merely the related item embeddings for model training. (2) Secure FedRec where users maintain the full model for training. It can be observed that SecEmb leads to substantial saving in memory and storage cost when the sparse item embedding matrix dominates the model parameters. For memory cost, the average savings are 12x, 21x, 101x, and 214x for ML1M, ML10M, ML25M, and Yelp, respectively. For storage cost, the average savings are 13x, 23x, 111x, and 247x for ML1M, ML10M, ML25M, and Yelp, respectively.

## K.3. Comparison with Sparse Aggregation Protocol

In this section, we discuss the advantages of our SecEmb over existing sparse aggregation protocols. We consider two SOTA frameworks, Secure Aggregation with Mask Sparsification (SecAggMask) (Liu et al., 2023) and Top-k Sparse Secure Aggregation (TopkSecAgg) (Lu et al., 2023). The key problem with the two frameworks is that they fail to ensure that the server learns nothing except the aggregated gradients. In particular:

- Leakage of rated item index. For SecAggMask, each user transmits the union of gradients with non-zero updates and masks to the server. For TopkSecAgg, each user is required to upload the coordinate set of non-zero gradients along with a small portion of perturbed coordinates. In both methods, the server could narrow down the potential rated items to a much smaller set.

- Leakage of gradient values. While TopkSecAgg protects the values of non-zero updates against the server, SecAggMask would reveal the plaintext values to the server. Specifically, SecAggMask randomly masks a portion of the gradients to reduce communication cost, and fails to ensure that all non-zero gradients would be masked against any attackers.

As the above sparse aggregation protocols focus on minimizing the transmission during update aggregation stage, we compare the upload communication between our SecEmb and these protocols in Table 13. For SecAggMask, we adopt a mask threshold such that 60% non-zero gradients would be masked in expectation. For TopkSecAgg, we set the perturbation proportion $\mu$ to be 0.1, following (Lu et al., 2023). Both approaches result in higher communication cost than SecEmb

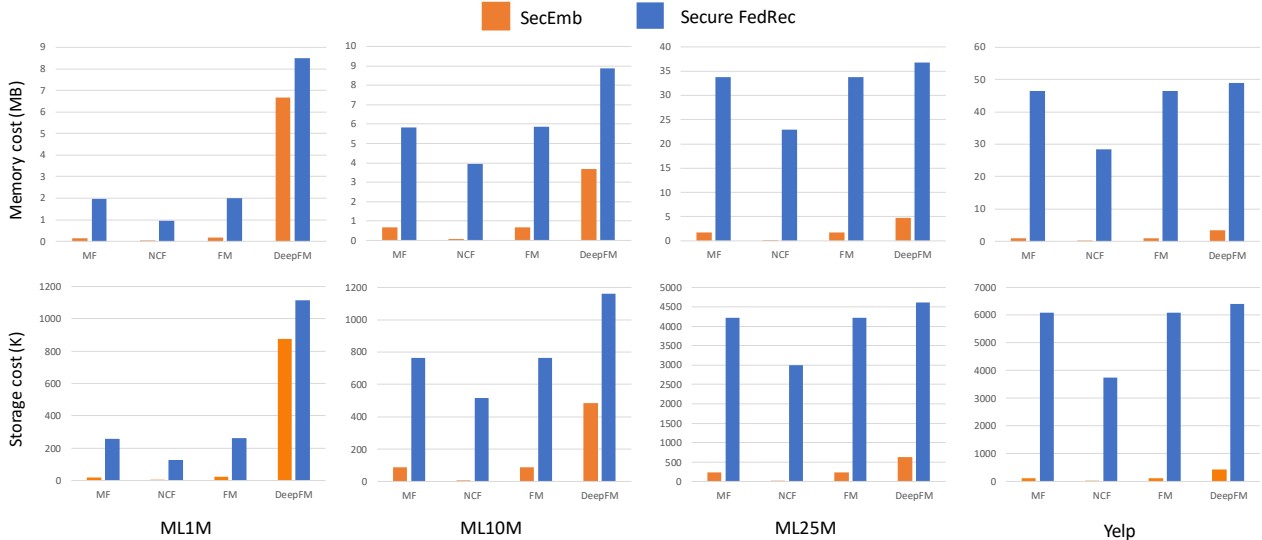

*Figure 8.* Average memory cost (in MB) and storage cost (in $10^3$ parameters) per user during training phase. The memory cost is computed with batch size of 1.

because: (1) Besides the non-zero embedding gradients, SecAggMask requires the user to send a certain proportion of randomly masked zero updates to the server. (2) To cancel out the mask values, in TopkSecAgg each user sends the union of rated item embeddings for all participating user, rather than the those for each single user.

*Table 13.* Communication cost (in MB) per user for SecEmb and Sparse SecAgg during upload transmission in one iteration. General SecAgg adopts two-server ASS that has minimum communication overhead.

|  |  | ML100K (1.7k Items) | ML1M (3.9k Items) | ML10M (10.7k Items) | ML25M (62.4k Items) | Yelp (93.4k Items) |
|---|---|---|---|---|---|---|
| MF | General SecAgg | 0.87 | 2.02 | 5.55 | 32.46 | 48.56 |
|  | SecAggMask | 0.27 | 0.61 | 1.66 | 9.60 | 14.35 |
|  | TopkSecAgg | 0.32 | 0.66 | 0.91 | 1.17 | 1.95 |
|  | Kvsagg | 1.21 | 2.49 | 3.41 | 4.37 | 7.31 |
|  | **SecEmb** | 0.17 | 0.27 | 0.28 | 0.51 | 0.52 |
| NCF | General SecAgg | 0.45 | 1.03 | 4.20 | 24.48 | 30.64 |
|  | SecAggMask | 0.14 | 0.31 | 1.25 | 7.21 | 8.98 |
|  | TopkSecAgg | 0.17 | 0.34 | 0.69 | 0.89 | 1.23 |
|  | Kvsagg | 0.61 | 1.25 | 2.57 | 3.29 | 4.58 |
|  | **SecEmb** | 0.13 | 0.20 | 0.26 | 0.46 | 0.43 |
| FM | General SecAgg | 0.93 | 2.04 | 5.56 | 32.47 | 48.57 |
|  | SecAggMask | 0.27 | 0.61 | 1.66 | 9.60 | 14.35 |
|  | TopkSecAgg | 0.32 | 0.66 | 0.91 | 1.17 | 1.95 |
|  | Kvsagg | 1.21 | 2.49 | 3.41 | 4.37 | 7.31 |
|  | **SecEmb** | 0.22 | 0.29 | 0.29 | 0.52 | 0.53 |
| DeepFM | General SecAgg | 14.96 | 8.87 | 8.72 | 35.63 | 51.20 |
|  | SecAggMask | 14.30 | 7.44 | 4.81 | 12.76 | 16.99 |
|  | TopkSecAgg | 14.36 | 7.49 | 4.07 | 4.32 | 4.58 |
|  | Kvsagg | 15.25 | 9.31 | 6.57 | 7.53 | 9.94 |
|  | **SecEmb** | 14.26 | 7.12 | 3.45 | 3.68 | 3.16 |

### K.4. Server Computation Cost

To validate the practicality of SecEmb, we evaluate the server computation cost under increasing number of devices during the training stage. Note that the server computation can be speed up utilizing the efficient full domain evaluation in (Boyle et al., 2016). Table 14 compares the computation time of our framework with homomorphic encryption (HE) approaches with CKKS cryptosystem (Cheon et al., 2017). Though both frameworks scales linearly with the number of participating devices, SecEmb is approximately 1600x faster than the typical HE protocol on average.

*Table 14.* Server computation cost (in minutes) per iteration for ML1M dataset.

| # of Active Users | 100 | 200 | 300 | 400 | 500 |
|---|---|---|---|---|---|
| HE (CKKS) | 127.26 | 244.51 | 391.76 | 519.05 | 646.38 |
| **SecEmb** | 0.08 | 0.16 | 0.25 | 0.33 | 0.41 |

### K.5. Breakpoint Analysis of Communication Cost

In Section 5.2.1 we show that SecEmb offers advantages over the secure FedRec in terms of upload communication as long as $m' < mbd/\left((\lambda+2)\log m + bd\right)$. The inequality usually holds for recommender system with sparse update. Note that the benefit on download communication is obvious where most users should rate over 50% of the total items to break the inequality—an unrealistic scenario in practice.

Table 15 presents the maximum number of $m'$ for each dataset where the aforementioned inequality holds. We use security parameter $\lambda = 128$ and 32-bit precision $b = 32$. It can be observed that the breakpoint of $m'$ is sufficiently large, over 50% of the total item size $m$. It is highly improbable for a user to rate such a substantial proportion of items in practical scenarios.

*Table 15.* Maximum Value of $m'$ for Communication Cost Advantage over secure FedRec under Various Embedding Dimension $d$.

| | ML100K (1.7k Items) | ML1M (3.9k Items) | ML10M (10.7k Items) | ML25M (62.4k Items) | Yelp (93.4k Items) |
|---|---|---|---|---|---|
| $d = 64$ | 1001 | 2210 | 5775 | 31038 | 45418 |
| $d = 128$ | 1255 | 2817 | 6408 | 37453 | 56031 |
| $d = 512$ | 1550 | 3547 | 9655 | 55418 | 82495 |

### K.6. Hyperparameters of Sequence Recommendation

We filter out users with less than 3 ratings in Amazon dataset (`https://cseweb.ucsd.edu/~jmcauley/datasets/amazon/links.html`), resulting in 9,267,503 items and 6,775,277 users. In each training round, 100 clients participate, and approximately 50,000 rounds are required to train the model on this extensive dataset. SecEmb and the four message compression baselines utilize the same hyperparameters as those used in prior sequence models. For SVD and CoLR, the rank of the reduced item embedding update matrix is set to 4 for the Amazon dataset and 8 for ML1M.

### K.7. Ablation Study

We compare SecEmb with two variants that eliminates one or two optimizations describe in Section 4.3. Table 16 and 17 present the per user upload communication cost and computation cost, respectively. The initial construction of SecEmb incurs over 20x higher communication and computation costs on average compared to the improved version, and the cost can be higher than that for secure FedRec on dataset with lower item size ($m <$11k). Furthermore, sharing the binary path between the two modules reduces the communication and computation cost by around 1.5x.

## L. Discussion

### L.1. Assumption of Two Non-Colluding Servers

Our protocol relies on two non-colluding servers for security. Note that user privacy is guaranteed as long as one of the servers is honest, even if the other colludes with any number of clients. In practice, the two servers can be: (1) a cloud service provider who makes the recommendation, and a third party who provides the cryptography or evaluation service; or (2) two third parties who provide the cryptography or evaluation service.

*Table 16.* Upload communication cost (in MB) for SecEmb and its variants.

|  |  | ML100K | ML1M | ML10M | ML25M | Yelp |
|---|---|---|---|---|---|---|
| MF | Secure FedRec | 0.86 | 1.99 | 5.47 | 31.96 | 47.81 |
|  | SecEmb-Init | 4.56 | 7.60 | 8.51 | 16.83 | 17.43 |
|  | SecEmb-RowEnc | 0.28 | 0.43 | 0.45 | 0.78 | 0.79 |
|  | SecEmb | 0.17 | 0.27 | 0.28 | 0.51 | 0.52 |
| NCF | Secure FedRec | 0.44 | 1.00 | 4.11 | 23.98 | 29.89 |
|  | SecEmb-Init | 2.32 | 3.86 | 6.42 | 12.70 | 11.00 |
|  | SecEmb-RowEnc | 0.18 | 0.28 | 0.38 | 0.67 | 0.61 |
|  | SecEmb | 0.13 | 0.20 | 0.26 | 0.46 | 0.44 |
| FM | Secure FedRec | 0.91 | 2.01 | 5.48 | 31.97 | 47.82 |
|  | SecEmb-Init | 4.80 | 7.84 | 8.74 | 17.16 | 17.47 |
|  | SecEmb-RowEnc | 0.29 | 0.44 | 0.46 | 0.80 | 0.79 |
|  | SecEmb | 0.18 | 0.28 | 0.29 | 0.53 | 0.53 |
| DeepFM | Secure FedRec | 14.95 | 8.84 | 8.63 | 35.13 | 50.45 |
|  | SecEmb-Init | 18.84 | 14.66 | 11.89 | 20.23 | 20.10 |
|  | SecEmb-RowEnc | 14.33 | 7.27 | 3.61 | 3.95 | 3.43 |
|  | SecEmb | 14.22 | 7.10 | 3.45 | 3.68 | 3.16 |

*Table 17.* Computation cost (in milliseconds) for secret generation per user for SecEmb and its variants.

|  |  | ML100K | ML1M | ML10M | ML25M | Yelp |
|---|---|---|---|---|---|---|
| MF | Secure FedRec | 0.79 | 1.80 | 4.91 | 28.67 | 51.04 |
|  | SecEmb-Init | 5.34 | 15.35 | 22.56 | 28.53 | 26.43 |
|  | SecEmb-RowEnc | 0.37 | 0.90 | 0.91 | 1.34 | 1.47 |
|  | SecEmb | 0.31 | 0.47 | 0.47 | 0.72 | 0.74 |
| NCF | Secure FedRec | 0.40 | 0.92 | 3.71 | 21.68 | 27.09 |
|  | SecEmb-Init | 5.50 | 8.76 | 15.03 | 25.95 | 21.76 |
|  | SecEmb-RowEnc | 0.57 | 0.65 | 0.70 | 1.27 | 1.15 |
|  | SecEmb | 0.37 | 0.44 | 0.48 | 0.81 | 0.80 |
| FM | Secure FedRec | 0.83 | 1.82 | 4.91 | 28.65 | 51.14 |
|  | SecEmb-Init | 18.30 | 19.04 | 29.02 | 29.28 | 37.95 |
|  | SecEmb-RowEnc | 0.69 | 0.75 | 0.70 | 1.07 | 1.15 |
|  | SecEmb | 0.46 | 0.50 | 0.48 | 0.74 | 0.74 |
| DeepFM | Secure FedRec | 13.16 | 7.84 | 7.69 | 38.84 | 53.90 |
|  | SecEmb-Init | 38.94 | 35.74 | 37.78 | 34.85 | 39.94 |
|  | SecEmb-RowEnc | 13.15 | 7.88 | 4.84 | 5.48 | 6.33 |
|  | SecEmb | 12.99 | 6.67 | 3.45 | 3.77 | 3.54 |

While two non-colluding parties is a common assumption in multi-party computation (MPC) protocols (Corrigan-Gibbs & Boneh, 2017; Addanki et al., 2022; Boneh et al., 2021; Mohassel & Zhang, 2017), it would be valuable to increase the colluding threshold of our protocol for enhanced security. A multi-party FSS for point function with full threshold $t > 2$ achieves a key size subpolynomial in domain size (Boyle et al., 2015; 2022), as opposed to the exponential reduction in the two-party case. For item embedding updates, the communication cost scales roughly with the square root of the item size $m$, rather than logarithmic in $m$ as in the two-party setting. A scalable solution with an increased collusion threshold requires a more communication-efficient scheme.

### L.2. Practical Considerations for Implementation of SecEmb

First, users might drop out during the implementation of our protocol. Our algorithm consists of two communication rounds per iteration: one for private embedding retrieval and another for secure update aggregation. Let $\mathcal{U}_1$ and $\mathcal{U}_2$ denote the participating users in these rounds, with $\mathcal{U}_2 \subseteq \mathcal{U}_1$. If users drop out in the update aggregation phase, servers could

simply aggregate the gradients from users in $\mathcal{U}_2$. Since each user independently generates their upload messages, the update aggregation on the retaining users remains unaffected.

Second, the aggregated gradients and updated models should be within the range $[-R, R]$ to avoid overflow in group operation. Consider $n_p$ participating users in each iteration, each user's input should be constrained to $[-R_u, R_u]$, where $R_u = (R-1)/n_p$. In SecEmb, this can be enforced by limiting the size of $CW^{(n+1)}$ to $\lfloor \log_2 R_u \rfloor \cdot d$ bits, thereby restricting the range of values each user can transmit.

Finally, we consider a dynamic setting where users rate new items during the training process. Since $m'$ is a pre-specified, unified value determined at the start based on the average $m'_u$, additionally ratings might increase each user's $m'_u$, potentially making $m'$ insufficient for most users. To address this issue, we periodically update $m'$ based on the average of $m'_u$. While a larger $m'$ increases user payload, the efficiency gains remain substantial as long as $m'$ is sufficiently small compared with $m$.

### L.3. Application to Language Model Training

Our framework optimizes training efficiency in the embedding layer, making it applicable beyond RecSys to models with sparse embeddings updates, such as language models (LM) (Dubey et al., 2024; Sanh, 2019; Chen et al., 2024b;a; 2025). In the federated training of LM, users retrieve related token embeddings via our private embedding retrieval protocol, update the model locally, and upload the gradients in our secure update aggregation module. Specifically, each user encodes the relevant token ids (i.e., token ids appear in their local dataset) with FSS keys, and the server computes secret shares of token embeddings for embedding retrieval or aggregates the secret shares of token embedding updates for SecAgg. For scalable federated LM training, it is essential to integrate our framework with other parameter-efficient finetuning methods (Han et al., 2024; Hu et al., 2022).

