# OpenReview forum: "SecEmb: Sparsity-Aware Secure Federated Learning of On-Device Recommender System with Large Embedding"
_ICML.cc/2025/Conference — ICML 2025 poster_

### Official Review · Reviewer_sPkh · 2025-03-07

**Overall Recommendation:** 3

**Summary:**

This paper proposes a privacy-preserving retrieval and aggregation method, SecEmb, that can be applied to federated recommender systems that leverage sparse embeddings. Likewise, due to the latent representation, SecEmb achieves faster computational & communication times versus standard (uncompressed) FedRec protocols. This is important in the federated setting, where communication, memory, and computational constraints are heavily present.

**Claims And Evidence:**

> Our SecEmb consists of two
correlated modules: (1) a privacy-preserving embedding
retrieval module that allows users to download relevant em-
beddings from the server, and (2) an update aggregation
module that securely aggregates updates at the server.

Section 4 nicely details each piece and provides insight into SecEmb's communication efficiency.

**Essential References Not Discussed:**

Not to my knowledge.\.

**Experimental Designs Or Analyses:**

I did not thoroughly check the validity of the experimental analysis.

**Methods And Evaluation Criteria:**

The paper does have a nice experimental section comparing SecEmb to many other compressed messaging schemes. It is a robust section which does showcase SecEmb's (marginal) improvement in performance.

**Other Comments Or Suggestions:**

I would add more information about the problem setting. I had to dig a bit further into your cited work (e.g., Xue 2017) to better understand the low-dimensional latent factor setting. Specifically, a better understanding of the mapping from X,Y to P,Q and then going from P,Q to the recommendation value. Furthermore, aren't R and Y related? I didn't see this mentioned.

**Other Strengths And Weaknesses:**

## Strengths
1. Effective retrieval and aggregation method applied to Federated RecSys. This is useful for the burgeoning federated recsys area.
2. Well-written.
3. Good empirical performance.

## Weakness
1. Slightly unclear RecSys problem formulation (see comment below).
2. The novelty simply revolves around applying a new secure retrieval and aggregation method fit for existing latent RecSys methods. Much of the speedup arises from using a smaller, compressed space.

**Questions For Authors:**

1. What are the compression rates used for the other message compression methods? Are the hyper-parameters consistent across the methods?
2. Is the novelty of this work simply constructing a secure message process on top of existing latent RecSys methodology?

**Relation To Broader Scientific Literature:**

It seems that this paper is slightly orthogonal to actual Federated RecSys training (i.e. actual compression) and instead focuses on how to transmit the compressed representations in a secure manner. I would ask for a more detailed description of the Federated RecSys setting (see my question below). However, I am not familiar enough in the secure messaging area to know how novel this new messaging scheme is.

**Theoretical Claims:**

I did not check the correctness. I will mention that the Theorems provided and proven in the supplementary material should be summarized in the main body.

---

> ### Author Rebuttal · Authors · 2025-04-01
>
> Thanks for your insightful comment and recognition of our work. Hope our response below could address your concerns.
>
> *Q1*: I will mention that the Theorems provided and proven in the supplementary material should be summarized in the main body.
>
> *Ans*: Thanks for your suggestion. We will move Theorem F.1 and F.2 to the main body to highlight our theoretical insights.
>
> *Q2*: Questions on problem setting – mapping from X,Y to P,Q and then going from P,Q to the recommendation value & relationship between R and Y.
>
> *Ans*: We will add more explanation in 3.1. Problem Statement to make the setting clearer. P and Q refer to the user and item embedding respectively, while X and Y refer to the user and item attributes respectively. P and Q are matrices that **encode the user or item ids into embeddings**. X and Y represents **attributes of users or items**, such as demographic details, genre, or price.
>
> For example, in DeepFM, we map the user id into a vector $p\in \mathbb{R}^d$ using matrix P, and map the item id into a vector $q\in \mathbb{R}^d$ using matrix Q. Similarly, X and Y are converted into embedding matrices $V_x\in \mathbb{R}^{l_x\times d}$ and $V_y\in \mathbb{R}^{l_y\times d}$, where $l_x$ and $l_y$ are the number of user and item attributes. Then the embeddings $p$, $q$, $V_x$, and $V_y$ are concatenated together and go through several hidden layers to obtain the final prediction.
>
> Below we further provide explanations of R, Y, and X:
>
> - R represents the ratings users assign to items (the item and user attributes are NOT included). $R_u$ is the collection of (item id, rating) for user u.
> - Y denotes item attributes, such as price and genre. $Y\in \mathbb{R}^{m\times l_y}$ gives the attributes for each item, and $y_i\in \mathbb{R}^{l_y}$ is the attributes for item i.
> - X is the attribute of users, such as their age. $X\in \mathbb{R}^{n\times l_x}$ gives the attributes for each user, and $x_u\in \mathbb{R}^{l_x}$ is the attributes for user u.
>
> Therefore, R, X, and Y provide distinct information, ensuring they complement rather than overlap with each other.
>
> *Q3*: The novelty simply revolves around applying a new secure retrieval and aggregation method fit for existing latent RecSys methods.
>
> *Ans*: Our work aims to construct a **lossless and efficient secure training protocol** for embedding-based RecSys. For security, we would like to ensure that the server learns nothing about individual gradients except the aggregated results during federated training. To achieve this goal, existing FL protocols typically adopt full model downloading and SecAgg, which incur significant communication and computation cost.
>
> This paper designs an efficient protocol **based on the properties of embedding-based RecSys**: 1) users typically interact with only a small fraction of available items; 2) only the interacted item embedding are relevant for prediction, and thus the gradient of item embedding is a sparse matrix. Accordingly, we design a secure and lossless protocol that is efficient both computationally and communicationally. **In Table 10 (Appendix J) we compare SecEmb with existing protocols, showing the distinct advantage of SecEmb in terms of efficiency, security, and utility for resource-constrained edge devices, highlighting our contribution of significant speedup while ensuring accuracy and security.**
>
> Note that **it is a non-trivial task to simultaneously ensure security and succinct user computation & communication cost** (i.e., cost depends linearly on the number of non-zero elements in each user’s update, and independent of or logarithmic in the total parameter size), and to the best of our knowledge, **there is no work that simultaneously achieves the two goals for embedding-based models (including LM and RecSys)**. We address the problem by a novel construction of FL training algorithm for embedding-based RecSys, which can be extended to the training of other large embedding models, such as language model (see Appendix L.2.  and response to Reviewer EqFa’s Q6).
>
> *Q4*: What are the compression rates used for the other message compression methods? Are the hyper-parameters consistent across the methods?
>
> *Ans*: The compression rates for other message compression methods, given by the Reduction Ratio in Table 2 Section 5, are computed as a result of:
>
> - For SVD and CoLR, based on the **rank** of the reduced item embedding update matrix (Table 9, Appendix I). The ranks are chosen to ensure a comparable reduction ratio with SecEmb while preserving utility.
> - For Bit8quant and Ternquant, based on the **bit size per value** of the quantized embedding udpates. Note that the bit size to represent a single value is fixed for each quantization method.
>
> The **hyper-parameters for SecEmb and other message compression methods remain consistent** across the models and datasets, using those given by Table 7 in Appendix I.

---

### Official Review · Reviewer_WwRR · 2025-03-11

**Overall Recommendation:** 3

**Summary:**

In the context of federated recommendation systems, this paper proposed the SecEmb protocol based on the Functional Secret Sharing algorithm and its coding property, which combines on-device request and update consistency on item indices. The protocol allows the client-side to only download and upload the embeddings of rated items, rather than the complete embedding table. It also ensures that the server is unaware of the rated items of each user and corresponding updated embeddings. Compared to the Secure FedRec, it significantly reduces the communication overhead between edge devices and the servers, as well as the computational overhead on the user side. Unlike existing dimensionality reduction and quantization methods, it retains complete information without sacrificing accuracy.

**Claims And Evidence:**

Yes.

**Essential References Not Discussed:**

The comparison with Kvsagg: Secure aggregation of distributed key-value sets published in ICDE 2023 should be added.

**Experimental Designs Or Analyses:**

Evaluation can be enhanced as follows:
- The criteria for dataset splitting should be added. Is it divided according to the chronological order of all samples, the rating time of each user, or some other method?
- More large-scale datasets are desired, such as the series of Amazon datasets.
- The current mainstream sequence recommendation models should be added for evaluation.

**Methods And Evaluation Criteria:**

The evaluation was conducted on MovieLens and Yelp datasets using factorization-based models.

**Other Comments Or Suggestions:**

The font size in most figures is somewhat small and difficult to read (for instance, the text in the overall framework of Figure 1 is not clear enough). The distinction between X, Y, and P, Q is unclear in Section 3, and the symbols x and i are somewhat confusing in terms of their meanings in Section 4.

**Other Strengths And Weaknesses:**

The key innovation of this work lies in effectively utilizing redundant data from two transmissions to minimize various overheads in federated learning as much as possible. However, a limitation of this method is requiring at least two server parties, and it must be ensured in the application that the data from these two parties can be collated. Other suggestions on evaluation and comparison with related work can be found above.

**Questions For Authors:**

Q1: Is it possible to adapt this method to a single server?

Q2: Compared to the method of downloading the full embedding table, does this approach result in any loss in recommendation effectiveness?

**Relation To Broader Scientific Literature:**

Federated learning for recommendation models with item embeddings.

**Theoretical Claims:**

Yes, I checked Section Complexity and Security Analysis.

---

> ### Author Rebuttal · Authors · 2025-04-01
>
> Thanks for your valuable comment and suggestions. Hope our response below could address your concerns.
>
> *Q1*: The criteria for dataset splitting should be added.
>
> *Ans*: We randomly split the ratings **for each user** into training and testing set.
>
> *Q2*: More large-scale datasets are desired & Add evaluation on sequence recommendation models
>
> *Ans*: We add experiments on the full Amazon datasets (https://cseweb.ucsd.edu/~jmcauley/datasets/amazon/links.html). After filtering out users with less than 3 ratings to train sequence recommendation model, we have 9,267,503 items and 6,775,277 users.
>
> We test two sequence models: Caser[1] and SASRec[2] (using hyperparameters in [1] and [2]), with results presented in Table R1. SecEmb reduces the upload communication cost by up to 2500x under huge item size (9 million+) and highly sparse data (density<0.002‰) while maintaining accuracy.
>
> [**Table R1.** Accuracy and Reduction Ratio (R.R.) using sequence models on Amazon dataset.]
> |||SecEmb|Bit8quant|Ternquant|SVD|CoLR|
> |-|-|-|-|-|-|-|
> |Caser|HR@10|0.629|0.628|0.623|0.627|0.625|
> ||NDCG@10|0.483|0.481|0.478|0.483|0.480|
> ||R.R.|2549|4|8|38|38|
> |SASRec|HR@10|0.639|0.636|0.635|0.621|0.600|
> ||NDCG@10|0.485|0.485|0.483|0.472|0.463|
> ||R.R.|2342|4|8|12|12|
>
> [1] Tang, J., & Wang, K. (2018). Personalized top-n sequential recommendation via convolutional sequence embedding. In Proceedings of the eleventh ACM international conference on web search and data mining (pp. 565-573)
>
> [2]Kang, W. C., & McAuley, J. (2018). Self-attentive sequential recommendation. In 2018 IEEE international conference on data mining (ICDM) (pp. 197-206).
>
> *Q3*: The comparison with Kvsagg should be added.
>
> *Ans*: We will add the comparison as follows:
>
> - **Kvsagg leaks more information to the server than SecEmb**. Beyond aggregated results, KvsAgg exposes the exact number of clients who rated each item within a batch, whereas SecEmb limits the server's knowledge to the aggregate alone.
> -  Kvsagg has a **minor failure rate**, occasionally yielding inaccurate aggregation results, whereas SecEmb ensures error-free aggregation.
> - **Kvsagg incurs higher communication costs than SecEmb.** Kvsagg's cost scales with the **union** of all participating users' rated items, whereas SecEmb's cost scales with each single user's rated items. Table R2 presents the upload communication cost of the two methods, showing SecEmb's communication benefits.
>
> [**Table R2.** Upload communication cost (in MB) using MF.]
> ||ML100K|ML1M|ML10M|ML25M|Yelp|
> |-|-|-|-|-|-|
> |Kvsagg|1.21|2.49|3.41|4.37|7.31|
> |SecEmb|0.17|0.27|0.28|0.51|0.52|
>
> *Q4*: Issues of two server setting
>
> *Ans*: **Two server setting is common in MPC protocols and has been adopted in industry by companies such as Goolge and Apple** (see Response to Reviewer TCtN's Q2). In SecEmb, one of the servers could be recommendation service provider, and the other could be a governmental organisation or a privacy service provider. Both parties have sufficient motivation to train a strong model and perform the computation correctly.
>
> Compared with two server protocols, the single server secure computation solutions generally incur significant overhead. They either require multiple rounds of communication and heavier user computation[3], or rely on computationally intensive homomorphic encryption[4]. While it becomes inefficient under a single server setting, we can relax the two non-colluding server assumption using a multi-party distributed point function (see Response to Reviewer TCtN's Q2).
>
> [3]Bonawitz, K., ... & Seth, K. (2017). Practical secure aggregation for privacy-preserving machine learning. In proceedings of the 2017 ACM SIGSAC Conference on Computer and Communications Security (pp. 1175-1191).
>
> [4]Chai, D., ... & Yang, Q. (2020). Secure federated matrix factorization. IEEE Intelligent Systems, 36(5), 11-20.
>
> *Q5*: The font size in most figures is somewhat small.
>
> *Ans*: Thanks for pointing out. We will refine the figures to enlarge the font size.
>
> *Q6*: The distinction between X, Y, and P, Q is unclear in Section 3, and the symbols x and i are somewhat confusing in terms of their meanings in Section 4.
>
> *Ans*: P and Q are embedding matrices that **encode the user or item ids into embeddings**. X and Y represents **attributes of users or items**, such as demographic details, genre, or price. (see Response to Reviewer sPkh's Q2)
>
> i denotes the target item id, while x denotes any item id fed into the point function. The point function outputs non-zero value only when x=i.
>
> *Q7*: Compared to downloading the full embedding table, does this approach result in any loss?
>
> *Ans*: SecEmb does NOT result in any loss compared with full embedding download, as long as users utilize only the embeddings of the target and previously interacted items to predict the rating or likelihood (the embeddings of other items are irrelevant). Most embedding-based RecSys models satisfy this criterion (see Response to Reviewer TCtN's Q1).

---

> > ### Comment · Reviewer_WwRR · 2025-04-09
> >
> > Thanks for your detailed response! After reading the response, the reviewer still has some key concerns as follows and thus will maintain the score.
> >
> > 1. What are the setups (e.g., how many clients each round, how many rounds to converge, the machine configurations) and the detailed time and communication overhead of SecEmb, Bit8quant, Ternquant, SVD, and CoLR for evaluating two sequence models on the full Amazon datasets? Implementation details should be included.
> >
> > 2. How the parameter m'_u affects the security/level analysis of the designed protocol? Since a small m'_u is the key of reducing the cost, but will it impact the security guarantee of the proposed protocol? The formal analysis on Page 16 misses the analysis on m'_u or the empirically set m' across all the users.
> >
> > 3. The design relies on private embedding retrieval for download and function secret sharing for upload, where the latter is a new application in federated recommendation. What is the new technical contribution of this work in terms of secure protocol design and analysis?

---

> > > ### Author Response · Authors · 2025-04-09
> > >
> > > Dear Reviewer WwRR,
> > >
> > > Thank you for your thoughtful follow-up. We hope the responses below address your concerns clearly and thoroughly.
> > >
> > > **Q1**: What are the setups (e.g., how many clients each round, how many rounds to converge, the machine configurations) and the detailed time and communication overhead of SecEmb, Bit8quant, Ternquant, SVD, and CoLR for evaluating two sequence models on the full Amazon datasets? Implementation details should be included.
> > >
> > > **Ans**: There are 100 client in each round, and it requires around 50000 rounds to train the model on the huge Amazon dataset. We use a server with L40 GPU and Intel Xeon Platinum 8336C CPU (CUDA version 12.2). The four baselines are integrated with the most efficient SecAgg protocol (2-server additive secret sharing) to ensure security. SecEmb and the four baselines adopt the same hyperparameters (in the original work of Caser and SASRec) for each sequence model. The rank of the reduced item embedding update matrix for SVD and CoLR is set as 4.
> > >
> > > The user computation time and communication overhead are listed in Table R1. It can be observed that **SecEmb results in communication cost reduction by at least 180x and computation cost reduction by at least 40x compared with the four baselines, since it effectively leverages the sparsity of the huge dataset for payload optimization**. We will detail the setting and overhead in our paper.
> > >
> > > [**Table R1.** Computation cost per user (in seconds) and upload communication cost (in MB) in one iteration on Amazon dataset.]
> > > |||SecEmb|Bit8quant|Ternquant|SVD|CoLR|
> > > |-|-|-|-|-|-|-|
> > > |Caser|Computation cost|0.011|0.527|0.518|0.870|0.451|
> > > ||Communication cost|1.36|866.65|433.33|276.65|276.65|
> > > |SASRec|Computation cost|0.004|0.324|0.316|0.443|0.285|
> > > ||Communication cost|1.48|864.21|432.27|276.77|276.77|
> > >
> > > **Q2**: How the parameter m'_u affects the security/level analysis of the designed protocol? Since a small m'_u is the key of reducing the cost, but will it impact the security guarantee of the proposed protocol? The formal analysis on Page 16 misses the analysis on m'_u or the empirically set m' across all the users.
> > >
> > > **Ans**: In Appendix C, we detailed the method to select a universal $m'$ for all users. The universal $m'$ is determined by the average of all users' individual $m_u'$, using a SecAgg protocol with only $O(1)$ communication and computation overheads per user. Therefore, **the server learns only the overall average rated item size, without gaining any information about individual $m'_u$ values**.
> > >
> > > **Since the $m'$ is pre-determined in advance, the value of $m'$ is unrelated to the security guarantee of the protocol.** Each user sends the same number of keys, concealing both their actual $m_u'$ and the corresponding key values and indices. Note that the formal analysis in Appendix F states that under **a pre-determined and universal $m'$**, the collection of $m'$ FSS keys hide **everything** about the user's gradient.
> > >
> > > **Q3**: The design relies on private embedding retrieval for download and function secret sharing for upload, where the latter is a new application in federated recommendation. What is the new technical contribution of this work in terms of secure protocol design and analysis?
> > >
> > > **Ans**: Thanks for your question. We would like to highlight our contributions as follows:
> > >
> > > - Existing FL protocols consider only the payload optimization for upload or download stage **separately**, while our method **cleverly leverage the connection between upload and download transmission** to optimize the computation and communication overhead. Specifically, we identify a crucial property of the FSS key, and observe that the indices of relevant embeddings remain the same across both stages. Based on these observations, we design a significantly smaller and more efficient FSS key.
> > >
> > > - Given the optimized FedRec protocol, **we provide a rigorous security proof that our optimized construction of secEmb maintains security**, i.e., the server learns nothing except the updated models. Note that the security proof for the cryptographic protocol is subtle and non-trivial, especially when we transmit **a condensed & optimized version of FSS key instead of the raw FSS key** in the upload stage, which necessitates a from-scratch security analysis.
> > >
> > > ----
> > >
> > > Overall, **we sincerely appreciate your thoughtful and constructive feedback. You insightful feedback helps make our work more thorough and robust.** We would be truly grateful if you could kindly take our further responses into consideration and reconsider your evaluation.
> > >
> > > Best Regards,
> > >
> > > Authors

---

### Official Review · Reviewer_EqFa · 2025-03-15

**Overall Recommendation:** 3

**Summary:**

This paper proposes a lossless and efficient federated recommendation training protocol to address the challenge of balancing efficiency and privacy in federated recommendation systems.

Additionally, it explores the use of row-wise sparsity in embedding matrices to optimize computational load.

Extensive experiments demonstrate the protocol's superior performance in terms of communication and computational efficiency.

**Claims And Evidence:**

This paper makes a convincing case to a certain extent. Extensive experiments demonstrate the protocol's superior performance in terms of communication and computational efficiency.

**Essential References Not Discussed:**

This paper primarily focuses on addressing efficiency and privacy issues in federated recommendation systems. However, some essential related works, such as FedMF [1] and LightFR [2], are missing. Specifically, FedMF introduces a homomorphic encryption technique to ensure privacy in federated recommendation systems, while LightFR leverages learning to hash to balance efficiency and privacy. The authors should conduct a more thorough review of the field and incorporate relevant literature to enhance the completeness of their work.


[1] Chai D, Wang L, Chen K, et al. Secure federated matrix factorization[J]. IEEE Intelligent Systems, 2020, 36(5): 11-20.

[2] Zhang H, Luo F, Wu J, et al. LightFR: Lightweight federated recommendation with privacy-preserving matrix factorization[J]. ACM Transactions on Information Systems, 2023, 41(4): 1-28.

**Experimental Designs Or Analyses:**

I examined the experimental setup and analysis and found that the dimensional settings of NCF vary across different datasets, whereas other backbone models, such as MF, FM, and DeepFM, maintain consistent dimensions across datasets. This inconsistency may lead to an unfair comparison.

**Methods And Evaluation Criteria:**

The method proposed in this paper is relatively reasonable, using a lossless and efficient aggregation protocol to enhance efficiency and privacy. However, the evaluation criterion for utility in this paper is relatively outdated. Specifically, this paper uses RMSE to assess the performance of the algorithm, while the current mainstream methods for verifying the performance of recommendations mainly include HR, NDCG, MRR, etc.

**Other Comments Or Suggestions:**

As far as I know, this paper merely utilized the behavioral interaction information within the recommendation system. On this basis, FM is equivalent to MF without the utilization of additional auxiliary information. Therefore, does the experiment of MF and FM exist redundancy?

**Other Strengths And Weaknesses:**

Strengths:

    1. The overall logic of this article is relatively clear.

    2.This paper employs a large number of experiments and analyses to verify the effectiveness of the proposed method.

Weaknesses:

    1. In the experimental part, there are some inconsistent settings for the experiments, which may lead to unfair comparisons.

    2. The relevant work needs to be further supplemented to improve the overall progress in the relevant field.

    3. The paper lacks effective analysis on certain experimental setups, such as why additional relevant experiments on language models are conducted and the necessity of MF and FM experiments.

**Questions For Authors:**

The core topic of this paper is to explore the privacy and efficiency of the embedding tables in recommendation systems. However, the appendix includes additional experiments related to the embedding tables of language models, which may lead to ambiguity in the topic. This is because the entities of Embedding in recommendation systems and those in language models are different. In recommendation systems, the objects are items, while in language models, the objects are tokens. Therefore, the author needs to provide an analysis of the necessity of conducting the related experiments.

**Relation To Broader Scientific Literature:**

This paper leverages the sparsity of embedding matrices to achieve a lossless and efficient federated recommendation training protocol, which is likely to attract widespread attention from researchers.

**Theoretical Claims:**

I have examined the theoretical contributions of the paper and found that its original theoretical innovations are relatively limited.

---

> ### Author Rebuttal · Authors · 2025-04-01
>
> Thanks for your insightful comment and valuable suggestions. Hope our response below could address your concerns.
>
> *Q1*: The evaluation criterion for utility in this paper is relatively outdated.
>
> *Ans*: Our paper focuses on the rating prediction task, and thus utilizes RMSE to measure the discrepancy between predicted and actual ratings. We have added experiments to evaluate the HR and NDCG, with a subset of results presented in Table R1, showing the superior performance of SecEmb over baselines. In response to *Reviewer WwRR's Q2*, we also present the experiment results in terms of HR and NDCG for sequence recommender system.
>
> [**Table R1.** Accuracy on ML1M using MF.]
> ||SecEmb|Bit8quant|Ternquant|SVD|CoLR|
> |-|-|-|-|-|-|
> |HR@10|0.593|0.593|0.592|0.591|0.590|
> |NDCG@10|0.337|0.335|0.333|0.335|0.331|
>
> *Q2*: The paper’s original theoretical innovations are relatively limited.
>
> *Ans*: We would like to highlight the theoretical contributions of our work as follows:
>
> - **Reduction of User Overhead:** Existing SecAgg protocols incur user communication and computation overhead linear in the item size $m$, while we theoretically show that our protocol achieves exponential reduction to $\log m$ (Section 4.4.1).
>
> - **Security analysis:** We provide a rigorous theoretical proof that our optimized construction of SecEmb maintains security, i.e., the server learns nothing except the updated models (Appendix F). We will move Theorem F.1 and F.2 to the main body to highlight our theoretical insights.
>
> To the best of our knowledge, we are the first to achieve both security and minimal user overhead for embedding-based models, demonstrating efficiency benefits both theoretically and practically, while remaining lossless in principle and practice.
>
> *Q3*: The dimensional settings of NCF vary across different datasets
>
> *Ans*: For NCF we adjusted the embedding sizes based on each dataset's characteristics, but **the comparison is fair as we use consistent hyperparameters (including dimension) for both SecEmb and the baselines**. To further validate our approach, we added experiments to set the embedding size for ML100K, ML1M, and Yelp as 24, matching that of ML10M and ML25M. The results, presented in Table R2, demonstrate similar observations that SecEmb maintains utility and reduce upload communication cost by up to 76x for NCF.
>
> [**Table R2.** RMSE and Reduction Ratio (R.R.) for NCF under d=24.]
> |||SecEmb|Bit8quant|Ternquant|SVD|CoLR|
> |-|-|-|-|-|-|-|
> |ML100K|RMSE|0.944|0.951|0.953|0.947|0.961|
> ||R.R.|4.27|3.90|7.56|4.62|4.75|
> |ML1M|RMSE|0.902|0.904|0.913|0.908|0.919|
> ||R.R.|6.30|3.96|7.80|5.94|6.01|
> |Yelp|RMSE|1.029|1.031|1.032|1.037|1.059|
> ||R.R.|76.54|4.00|7.99|12.21|12.22|
>
> *Q4*: Some essential related works, such as FedMF and LightFR, are missing.
>
> *Ans*: We will add more discussion of both works in our paper as follows:
>
> - FedMF ensures privacy with HE techniques. Compared with SecEmb, FedMF incurs substantial computation overhead, and fails to simultaneously reduce user computation & communication cost and ensure security.
>
> - LightFR improves the efficiency by binarizing continuous user/item embeddings through learning-to-hash, but it suffers utility loss in principle and incurs overhead linear in item size m (as opposed to SecEmb's exponential reduction).
>
> *Q5*: Does the experiment of MF and FM exist redundancy?
>
> *Ans*: In our framework, MF updates involve only item embeddings, while FM updates encompass both item embeddings and item feature embeddings. This distinction results in differing parameter sparsities between the two models since item feature embeddings are not generally sparse (see Table 8, Appendix I). **By evaluating SecEmb's efficiency and utility across models with varying item embedding proportions and thus update sparsity ratios, we provide a thorough assessment of its performance.**
>
> *Q6*: Necessity of conducting the related experiments on language models
>
> *Ans*: We aim to show that **our framework is applicable to the federated training of language models by substituting rated item IDs with target token IDs in SecEmb's algorithm**. Roughly speaking, each user encodes the relevant token ids (i.e., token ids appear in their local dataset) with FSS keys, and the server computes secret shares of token embeddings for embedding retrieval or aggregates the secret shares of token embedding updates for SecAgg. It helps to reduce the overhead since the tokens appearing in each user’s local dataset are typically a small subset of the entire vocabulary. Table 17 shows that SecEmb achieves an overhead reduction ratio between 1.06 and 1.55 for full-parameter fine-tuning, and it can achieve higher reduction ratio when combining with parameter-efficient finetuning (e.g., up to 30x reduction when applying LoRA on Llama3-8b).

---

### Official Review · Reviewer_TCtN · 2025-03-17

**Overall Recommendation:** 3

**Summary:**

This paper introduces a secure federated recommender system, tailored to the case where user data is sparsely presented. Conventional federated secure aggregation methods suffer from unnecessary communication overhead. Thus, the authors use function secure sharing and propose two modules (embedding retrieval module and update aggregation module) that forms the proposed lossless secure protocol SecEmb. The computation and communication complexities are analyzed. Emprical results demonstrates the effectiveness of the proposed method.

**Claims And Evidence:**

Line 215: "only the embeddings for interacted items are relevant for model updates." Could the authors elaborate more on this claim? For example, is it a claim made for a specific type of recommendation systems, or it stands for any recommendation systems?

Other main claims seem to be supported by both complexity analysis and empirical experiments.

**Essential References Not Discussed:**

N/A

**Experimental Designs Or Analyses:**

Experiment design is sound.

**Methods And Evaluation Criteria:**

Yes

**Other Comments Or Suggestions:**

* Line 123. The introduction of l_x is abrupt, and its explanation only comes a few paragraphs later.
* The explanation for "remaining parameters theta" is vague

**Other Strengths And Weaknesses:**

The paper is overall well written, and the evaluation is comprehensive in the given scope. That being said, the scope is limited: the proposed method only applies to secure federated recommendation system, where user data is sparse, and with two non-colluding servers. The paper would be strengthened if it can be applied to a wider range of settings, such as only one trusted server, or where the sparsity is unknown.

**Questions For Authors:**

* Is it easy to relax the assumption of two non-colluding servers?
* How robust is the proposed method to local devices dropping out during training?

**Relation To Broader Scientific Literature:**

N/A

**Theoretical Claims:**

N/A

---

> ### Author Rebuttal · Authors · 2025-04-01
>
> Thanks for your insightful comment and positive rating of our paper. Hope our response below could address your concerns.
>
> *Q1*: Elaboration on "only the embeddings for interacted items are relevant for model updates"
>
> *Ans*: It works for recommender systems (RS) based on item embeddings, as long as **users utilize only the embeddings of the target and previously interacted items to predict the rating or likelihood (the embeddings of other items are irrelevant)**. In other words, to predict $r_{ui}$, user $u$ only utilizes the embeddings of item $i$ or previously interacted items $I_u$. Most embedding-based RSs satisfy this criterion, including but not limited to:
>
> - Latent factor-based collaborative filtering RSs and their extensions, such as MF, NCF, FM, and DeepFM.
> - Sequential RSs based on item embeddings, such as CNN-based models (Caser), RNN-based models (GRU4Rec), and Attention-based models (SASRec).
>
> *Q2*: Assumption on two non-colluding servers.
>
> *Ans*: **Two non-colluding server is a common assumption in MPC protocols [1][2][3], and has been adopted in industry.** For example, Prio[1] has been used by Google and Apple to measure the effectiveness of Covid-19 exposure-notification apps on iOS and Android[4] and improve the iOS Photos app[5]. In SecEmb, one of the servers could be recommendation service provider, and the other could be a governmental organisation or privacy service provider.
>
> Compared with two server protocols, the single server secure computation solutions generally incur significantly higher communication and computation cost. They either require multiple rounds of communication and heavier user computation[6], or rely on computationally intensive homomorphic encryption[7].
>
> **The two non-colluding servers assumption can be relaxed using m-party distributed point function (m>2), allowing collusion among up to m-1 parties**[8]. Each user generates m FSS keys for their updates, which are distributed to m servers for aggregation and reconstruction. However, this protocol has less overhead reduction than the two server setting, since user overhead depends sub-polynomially in item size $O(\sqrt{m})$, rather than logarithmically $O(\log m)$ in the two server case. We leave further reduction to future research.
>
> [1]Corrigan-Gibbs, H., & Boneh, D. (2017). Prio: Private, robust, and scalable computation of aggregate statistics. In 14th USENIX symposium on networked systems design and implementation (pp. 259-282).
>
> [2]Mohassel, P., & Zhang, Y. (2017). Secureml: A system for scalable privacy-preserving machine learning. In 2017 IEEE symposium on security and privacy (SP) (pp. 19-38).
>
> [3]Boneh, D., ... & Ishai, Y. (2021). Lightweight techniques for private heavy hitters. In 2021 IEEE Symposium on Security and Privacy (SP) (pp. 762-776).
>
> [4]Apple and Google. (2021). Exposure Notification Privacy-preserving Analytics (ENPA) white paper.
>
> [5]https://machinelearning.apple.com/research/scenes-differential-privacy
>
> [6]Bonawitz, K., ... & Seth, K. (2017). Practical secure aggregation for privacy-preserving machine learning. In proceedings of the 2017 ACM SIGSAC Conference on Computer and Communications Security (pp. 1175-1191).
>
> [7]Chai, D., ... & Yang, Q. (2020). Secure federated matrix factorization. IEEE Intelligent Systems, 36(5), 11-20.
>
> [8]Boyle, E., Gilboa, N., & Ishai, Y. (2015). Function secret sharing. In Annual international conference on the theory and applications of cryptographic techniques (pp. 337-367).
>
> *Q3*: Application to unknown sparsity.
>
> *Ans*: **SecEmb supports unknown sparsity, where the server lacks prior knowledge of the sparsity ratio**. Each user uploads a self-defined number of FSS keys based on their own data sparsity, which can exceed the number of interacted items to obscure it. The server aggregation still proceeds correctly with varying FSS key counts across clients. Appendix K.5. shows that SecEmb effectively reduces overhead when the density of item embedding update is within 50\% in general cases. This approach extends to other sparse embedding update settings (with unknown sparsity ratio), such as language model.
>
> *Q4*: Abrupt introduction of l_x & Vague explanation for "remaining parameters theta"
>
> *Ans*: Thanks for pointing out and we will revise the paper accordingly. $l_x$ is the number of user features. Remaining parameters $\theta$ refer to the parameters other than user and item embeddings. For example, $\theta$ in DeepFM is the collection of: 1) parameters in hidden and output layers, and 2) embeddings for user and item attributes (rather than user or item ids).
>
> *Q5*: Robustness to local devices drop out.
>
> *Ans*: For n participants, SecEmb is robust to up to n-1 dropouts. It has one communication round for private embedding retrieval and another round for secure update aggregation. If users drop out in the second round, servers could still aggregate the gradients for remaining users correctly (see Appendix L.3).

---

### Decision · Program_Chairs · 2025-05-01

**Decision:**

Accept (poster)

**Comment:**

This paper proposes a lossless secure recommender systems on sparse embedding updates (SecEmb), which reduces user payload while ensuring that the server learns no information about both rated item indices and individual updates except the aggregated model. All reviewers are in favor of weakly accepting this paper and authors have addressed most of the concerns during the rebuttal phase, so I recommend acceptance.